# Simulating ice thickness and velocity evolution of Upernavik Isstrøm 1849–2012 by forcing prescribed terminus positions in ISSM

Konstanze Haubner[1,2], Jason E. Box[1], Nicole J. Schlegel[3,4], Eric Y. Larour[3], Mathieu Morlighem[5], Anne M. Solgaard[1], Kristian K. Kjeldsen[6], Signe H. Larsen[1,7], Eric Rignot[3,5], Todd K. Dupont[8], and Kurt H. Kjær[2]

[1]Geological Survey of Denmark and Greenland (GEUS), Copenhagen, Denmark
[2]Centre for GeoGenetics, Natural History Museum, University of Copenhagen, Copenhagen, Denmark
[3]Jet Propulsion Laboratory (JPL), California Institute of Technology, Pasadena, CA, USA
[4]University of California, Los Angeles, CA, USA
[5]Department of Earth System Science, University of California-Irvine, Irvine, CA, USA
[6]DTU Space - National Space Institute, Technical University of Denmark, Department of Geodesy, Kgs. Lyngby, Denmark.
[7]Centre for Ice and Climate, Niels Bohr Institute, University of Copenhagen, Copenhagen, Denmark
[8]Miami University, Oxford, OH, United States

**Correspondence:** Konstanze Haubner (khu@geus.dk)

**Abstract.** Tidewater glacier velocity and mass balance are known to be highly responsive to terminus position change. Yet, it remains challenging for ice flow models to reproduce observed ice margin changes. Here, using the Ice Sheet System Model (ISSM; Larour et al., 2012), we simulate the ice velocity and thickness changes of Upernavik Isstrøm (NW Greenland) by prescribing a collection of 27 observed terminus positions spanning 164 years (1849–2012). The simulation shows increased ice velocity during the 1930s, the late 1970s and between 1995 and 2012 when terminus retreat was observed along with negative surface mass balance anomalies. Three distinct mass balance states are evident in the reconstruction: (1849–1932) with near zero mass balance, (1932–1992) with ice mass loss dominated by ice dynamical flow, and (1998–2012), when increased retreat and negative surface mass balance anomalies lead to mass loss twice that of any earlier period. Over the multi-decadal simulation, mass loss was dominated by thinning and acceleration responsible for 70 % of the total mass loss induced by prescribed change in terminus position. The remaining 30 % of the total ice mass loss resulted directly from prescribed terminus retreat and decreasing surface mass balance. Although the method can not explain the cause of glacier retreat, it enables the reconstruction of ice flow and geometry during 1849–2012. Given annual or seasonal observed terminus front positions, the method could be a useful tool for evaluating simulations investigating the effect of calving laws.

## 1 Introduction

In recent decades, glaciers terminating into the ocean (tidewater glaciers) have exhibited widespread thinning and velocity acceleration (e.g. Pritchard et al., 2009; Rignot et al., 2011; Velicogna et al., 2014; Khan et al., 2015). Increased air and ocean temperatures induce increased surface melt rates and frontal retreat (Podrasky et al., 2012; Rosenau et al., 2013; Moon et al., 2014), represented by submarine melt and iceberg calving. The Greenland ice sheet has more than 240 tidewater glacier

outlets (Rignot and Mouginot, 2012) and its mass balance is highly affected by changes in tidewater glacier discharge (van den Broeke et al., 2009; Bevan et al., 2012; McMillan et al., 2016). Greenland's ice mass changes have dominated global sea level contributions of the past two decades (e.g. Rignot et al., 2011; Gardner et al., 2013). Sea level projections rely on models to estimate discharge and Greenland's contribution to sea level that are coming more into line with observations (Shepherd and Nowicki, 2017). However, accurate simulation of terminus position remains a major challenge (Nick et al., 2009, IPCC, 2013 chapter 13).

Tidewater glacier retreat occurs due to calving (Benn et al., 2007; Nick et al., 2010) and submarine melt (Motyka et al., 2011; O'Leary and Christoffersen, 2013; Morlighem et al., 2016; Rignot et al., 2016). Yet, no universal calving law exists (Benn et al., 2007) and model approaches either (1) focus on the development and performance of a particular calving law (e.g. Cook et al., 2014; Todd and Christoffersen, 2014); (2) simplify glacier simulations using flow line or flow band models (e.g. Nick et al., 2013; Lea et al., 2014), neglecting e.g. across-flow stresses or (3) determine glacier terminus changes on ice particle scale and are thereby not well suited for long-term studies (Åström et al., 2013, 2014).

Upernavik Isstrøm (UI), a set of West Greenland tidewater glaciers, has been the focus of several observational studies. Weidick (1958) compiled historical records of UI terminus positions between 1849 and 1953, concluding that terminus retreat had increased starting in the 1930s. Observed periods of increased UI terminus retreat in 1931 to 1946, in the late 1990s and in 2005–2009 correlate with elevated air temperatures (Andresen et al., 2014). Two periods of increased dynamically driven ice loss that took place in 1985–1993 and 2005–2010 were responsible for 79% of the ice mass loss during 1985–2010 (Kjær et al., 2012; Khan et al., 2013).

Previous studies either simulate tidewater glacier retreat with ice flow models or discuss observed terminus changes and its implications for tidewater glaciers. In this study, we combine observations and ice flow models by using observed terminus positions in the Ice Sheet System Model (ISSM; Larour et al., 2012) to simulate Upernavik's glacial system evolution from 1849, near the end of the Little Ice Age, to 2012. We reconstruct the 1849 ice surface elevation and force ISSM glacier terminus retreat with 27 observed terminus positions.

This study does not aim to simulate physically caused retreat, instead we evaluate the effects of changing termini on UI's ice surface elevation and velocity. We (1) investigate whether prescribed terminus change produces a realistic thinning and velocity history; (2) compare simulated mass loss, surface elevation and velocity changes with 1985–2012 observations; and (3) correlate the calculated dynamic ice loss with observational studies. ISSM produces a weekly reconstruction of UI ice thickness and surface velocity from 1849–2012.

## 2 Area and data

UI has a catchment area of ~64,700 km$^2$, terminating into several tidewater glaciers. We focus on the three main glaciers and denote them UI-1, UI-2 and UI-3 from north to south (Fig. 1). Historically, the three glaciers shared the same terminus between 1849 and 1931 (Fig. 1; Weidick, 1958). In the 1930s the glaciers separated in two, UI-1/UI-2 and UI-3. UI-1 and UI-2 decoupled after 1966. Historical front positions (Fig. 1) were collected from several sources: 1849–1953 (historical records; Weidick,

1958), 1966–1975 (satellite images; Andresen et al., 2014), 1985–1996 (aerial photographs (1985) and satellite images; Khan et al., 2013), 1999 to 2012 (satellite images; Jensen et al., 2016).

For initialisation and evaluation of the model we use data from different studies, described in Table 1.

## 3   Ice flow model

We use the Ice Sheet System Model (ISSM; Larour et al., 2012), a finite-element thermomechanical ice flow model. Ice flow is calculated applying the Shelfy Stream Approximation (SSA; MacAyeal, 1989), that integrates vertically averaged ice properties (e.g. ice rheology, thickness, velocity) and neglects vertical shear stresses. The SSA is well suited for fast-flowing glaciers like Upernavik, where the ice flow is primarily driven by basal sliding.

Ice viscosity follows Glen's Flow law (Glen, 1955). The initial viscosity is taken from Table 3.4 in Cuffey and Paterson (2010, p. 75), assuming ice temperature of $-5$°C and will be refined in section 3.1.

A Budd-like friction law (Budd et al., 1979) is applied on all grounded ice:

$$\tau_b = -C^2 N v_b \tag{1}$$

where $v_b$ is the basal velocity. The effective pressure $N$ is assumed to be $N = g\left(\rho_i H(x,y,t) + \rho_w b(x,y)\right)$ where $H$ is the ice thickness at the current time step, $b$ is the bed elevation with respect to sea level, $g$ is the gravity, $\rho_i$ and $\rho_w$ are the densities of ice and water respectively. The friction coefficient $C$ is variable in space, but constant in time (Fig 1, supplementary). Perfect sliding is assumed on floating ice.

The model domain is set to the Upernavik catchment, which is defined by the flow direction given by the 2008/09 surface velocity from Rignot and Mouginot (2012) (red area in Fig. 1). We use an adaptive mesh that has a resolution varying between 300–800 m in the area of observed terminus changes and 12 km near the ice divide, resulting in about 17,000 mesh elements. Resolution increases with larger changes in ice velocity (Rignot and Mouginot, 2012) or bedrock topography (Morlighem et al., 2017) and decreases stepwise with distance from the front.

We impose hydrostatic pressure at the terminus and keep the ice velocity and surface elevation constant at the inland boundary. No submarine frontal melt or calving rates are applied, since the study aims to simulate ice velocity and thickness changes caused by observational prescribed terminus changes. The ice is allowed to float depending on a hydrostatic criterion (Seroussi et al., 2014).

### 3.1   Model initialisation

Since starting the simulation in 1849 extends the present day ice extent by 356 km², model initialisation requires reconstruction of the ice surface elevation in the extended area. To initialise the model we thus reconstruct the 1849 ice surface elevation, as described in the following. Over the present day ice covered area, the initial ice surface is given by the 2005 ice surface elevation (GIMP; Howat et al., 2014). At the 1849 marine terminus (given by  Weidick, 1958), the ice surface elevation is

set to 70 m a.s.l. consistent with marine termini in the area, based on IceBridge data (Krabill, 2010, updated 2016). Trimline data points (Fig. 1; Kjeldsen et al., 2015) mark the 1849 surface elevation and ice extent on the bedrock along the fjords. In the remaining area the ice surface elevation is interpolated linearly being constraint to a minimum elevation of 40 m. The ice thickness is set to floatation height or to the maximum thickness, defined through the initialised ice surface elevation and bed topography. The ice surface velocity is resolved performing a stress balance solution.

During the relaxation, the reconstructed glacier area thickened by 50–400 m from the UI-1/UI-2 1966 terminus position reaching 40 km upstream, while the ice surface velocity slowed down by a maximum of 2.5 km y$^{-1}$. Along UI-3, the glacier thickness changed by $\pm$150 m and ice surface velocity decreased by 500–1500 m y$^{-1}$ (see supplementary).

As we are interested in determining how the model geometry and velocity react to the prescribed terminus change and not internal model instability, we relax the model prior to the transient run, bringing ice surface elevation and velocity into equilibrium (following Schlegel et al. (2016)). Equilibrating model geometry and velocity requires constant forcing, i.e. a stable SMB. The SMB at Upernavik is found to be stable in 1854–1900 and 1964–1990. The mean 1854–1900 SMB value is used for equilibrating the model for 1849 conditions and 1964–1990 is set as the SMB reference period to evaluate simulated mass balance.

We perform two relaxation runs stepwise (Table 2), keeping SMB constant to the 1854–1900 mean value (Box, 2013). The first relaxation provides reconstructed 1849 ice thickness, given the GIMP surface elevation extended to the 1849 terminus. Thus, in the first relaxation basal friction is based on the assumption that driving stress is equal to basal stress at any given point using the initial geometry.

Given computed ice velocity and thickness from the first relaxation, ice viscosity and basal friction can be redefined. The ice viscosity is calculated by extruding the model with 15 layers and solving for the thermal steady state based on forcing the surface with 1854–1900 UI mean surface air temperature (Box, 2013). The basal friction coefficient is constant in time, but varies in space, and is calculated by an adjoint-based inversion, following Morlighem et al. (2010) and MacAyeal (1993), given the updated ice viscosity from the thermal steady state simulation.

The second relaxation runs for 5,000 years until ice velocity and thickness are equilibrated, provided with ice thickness from the first initialisation, simulated ice viscosity and inverted basal friction. The end state of this relaxation provides the initial values of simulated ice surface elevation and surface velocity for the 1849–2012 simulations.

## 3.2 Simulation setup

We run two different model simulations: (1) a control run ISSM$_{control}$, forced only by monthly SMB (Box, 2013) using a fixed terminus at the observed 1849 ice margin and (2) a prescribed terminus change simulation ISSM$_{PT}$, forced by the same monthly SMB and observed calving front positions. ISSM$_{control}$ serves to estimate the ice mass, velocity and thickness changes that are simulated without prescribed terminus change.

The prescribed terminus position change in ISSM$_{PT}$ is implemented through a levelset-based method (Bondzio et al., 2016, 2017) and performed in July of the observation year, according to observed terminus positions (Fig. 1). The highest surface air temperatures and melt rates on UI are observed in July (van As et al., 2016), increasing the likelihood of terminus retreat

(Sciascia et al., 2013; Fried et al., 2015). We introduce 20 additional calving front positions, created through linear interpolation between the observed termini positions and constrained by the mesh resolution. The additional calving fronts are prescribed at the halfway points in time between observations and aim to improve realistic simulation behaviour by splitting 20 large ice area changes induced by the prescribed terminus changes into smaller areas within shorter time periods (dotted lines, Fig 3).

Within the prescribed ice area, the grounding line is evolving freely and floating tongue formation is thereby allowed.
The simulation evaluation time step is set to 73 h, constrained by the Courant-Friedrich-Lewy condition (Courant et al., 1967), ensuring the numerical stability solving the ice flow equations at each time step.

## 4  Results and comparison

During the simulation, most of the ice thickness and velocity changes occur near the central flow lines of UI-1, UI-2 and
UI-3. Simulated changes in ice thickness and velocity in the majority of the model domain (more than 70 km inland from the 2012 terminus or 5 km away from the central flow lines of the three glaciers) are below 5 %, corresponding to changes of 20 m and 10 m y$^{-1}$ over 164 simulation years. Hence, in the following we present relative and absolute changes in ice velocity and thickness along the central flow lines of UI-1, UI-2 and UI-3 from the 2012 terminus reaching 30 km upstream (Fig. 1).

### 4.1  Model comparison

Between 1849 and 2012, ISSM$_{control}$ shows less than 7 % thinning and 5 % acceleration, simulating a change in velocity less than 120 m y$^{-1}$ and a thinning less than 30 m along the central flow lines for the entire period. In contrast, ISSM$_{PT}$ produces a thinning between 20 % along the flow lines and up to 60 % in the area between 2012 terminus and 70 km upstream in 1849–2012, corresponding to thinning between 100 and 450 m along the flow lines. The average ice surface velocity increase along UI-1 and UI-2 is 180 % and 47 % on UI-3. Cumulative ice mass loss over the simulation period of the entire model domain
(converted from modelled water equivalent assuming 917 kg m$^{-3}$ ice density) was by the end of the model simulation −50 Gt for ISSM$_{control}$ and −585 Gt for ISSM$_{PT}$ (Fig. 2). 99 % of simulated ISSM$_{control}$ mass loss was prescribed by SMB anomalies while 30 % of total ice mass loss simulated by ISSM$_{PT}$ was prescribed, with SMB anomalies accounting for 9 % (−50 Gt) and prescribed terminus position change contributed 21 % (−121 Gt). Thus, 70 % of by ISSM$_{PT}$ simulated mass loss is caused by thinning and acceleration. The following subsections describe ISSM$_{PT}$ results in more detail.

### 4.2  Mass balance

In the following section we focus on the simulated mass balance (MB) through the model runs (see cumulated mass change in Fig. 2). For marine terminating glaciers, mass balance can be attributed to either changes in SMB or changes in dynamic ice loss (DIL). A tidewater glacier is in equilibrium, when SMB and DIL are in balance. Deviations in SMB and DIL change the
glacier and its stability hereafter referred to as anomalies ∆SMB and ∆DIL. SMB is a model input and ∆SMB are calculated relative to the mean value of the stable UI period 1964–1990 SMB. ∆DIL is calculated as the residual between the simulated

MB and $\Delta$SMB.

The simulated annual MB for the UI catchment (Fig. 2) is positive from 1849 to 1920. In this period, the MB from the ISSM$_{PT}$ and ISSM$_{control}$ are similar due to very few and small terminus changes (Fig. 2) and MB is thus dominated by $\Delta$SMB. Anomalies in DIL (Fig. 3) are evident by small ($-0.5$ to $-4$ Gt) peaks that coincide with prescribed terminus retreat. After 1920, the MB becomes negative, except in 1996, when $\Delta$SMB has a peak (8 Gt), which is attributed to a high winter accumulation (McConnell et al., 2001; Box et al., 2006). Figure 3 highlights three periods in MB trends: (1) 1849–1932, when MB is near equilibrium, (2) from 1932 to 1992, when the negative MB is driven by $\Delta$DIL, and (3) 1998–2012, when SMB and DIL both have high negative anomalies and the total mass loss each year was twice as high as any year before.

Khan et al. (2013) and Larsen et al. (2016) measure surface elevation changes from aerial photographs, satellites and digital elevation models between 1985 and 2010. These yield a total mass change during different time periods and congruent to our calculations $\Delta$DIL is estimated as the residual of mass change and $\Delta$SMB. Both studies refer to different areas within the UI catchment. Table 3 presents a comparison of the observed mass changes and our simulation results, recalculated for the particular areas. Due to sparse data coverage, Khan et al. (2013) combine surface elevation measurements acquired between 2002 and 2005 to quantify elevation changes and refer to this period as 2002/2005. The average of simulated ice mass loss between 2002 and 2005 is taken for comparison with the 2002/2005 observations from Khan et al. (2013).

Simulated total ice mass changes in 1985–2002/2005 and 2006–2011 correspond with observed ice mass changes from Khan et al. (2013) and Larsen et al. (2016) (Table 3). Additionally, the DIL during 2000 to 2005 makes up 83 % of the mass change and in 2006–2011 this percentage is reduced to 64 %, in agreement with Khan et al. (2013) and Larsen et al. (2016). In 2000–2005, however, simulated total mass changes are 81 % larger than the maximum of what is observed.

A comparison with GRACE, that measures gravity field variations from which mass change is computed, shows equivalent seasonal mass loss fluctuations in summer and mass gain in winter with an overall negative trend. The simulated mass change rates resemble 98 % of GRACE's rate (see supplementary).

## 4.3 Ice thickness

ISSM$_{PT}$ simulates 10–80 % thinning from 1849 to 2012 over an area reaching 70 km upstream from the 2012 terminus (see supplementary). Transient surface elevation changes along the central flow lines of UI-1, UI-2 and UI-3 are visualised in the supplementary Fig. 8 and movie01 (supplementary). The model simulation shows increased surface lowering in the time periods 1930/40, 1970/80 and from 2000 onwards.

To evaluate simulated ice thickness, we compare simulation results with the residual ice thickness obtained from observed surface elevation data and the bed topography from Morlighem et al. (2017), that is used in the simulation setup. We refer to the supplementary for illustrations of spatial comparisons between simulation results and observations. Simulated thickness of the UI catchment in 2005 lies within $\pm 30$ % of observed thickness (GIMP), except in the shear margin regions of UI-1, where simulated ice thickness is too high by up to 160 % of observations. A comparison of absolute ice thickness in 2005 shows up to 200 m lower simulated thickness than observed, apart from the shear zones of UI-1, where the ice is up to 200 m thicker than observed. Differences between the simulated 2012 ice thickness and observations (ArcticDEM) show the same pattern with

less difference in the UI-1 shear zone. The 1985 DEM based on aerial photographs (Korsgaard et al., 2016) covers only the UI coastal area, reaching at most 40 km inland and covering primarily the UI-3 area. Simulated ice is 20 to 100 % thicker around UI-1 and UI-2 than the 1985 observations and 10 % thinner on average along UI-3.

NASA Operation IceBridge (Thomas and Studinger, 2010; Krabill, 2010, updated 2016) provides ice surface elevation along
UI-1 (2009–2012) and UI-3 (1994, 1999, 2002, 2009, 2010, 2012). A mean value comparison along the UI-3 flow line illustrates that the simulated ice thickness is on average 10 % less than observations (Fig. 4). The same comparison on UI-1 shows simulated ice thickness being 104 % of observations close to the UI-1 terminus and 93 % of observed thickness 5 to 10 km upstream the 2012 terminus. Observed IceBridge and simulated surface elevation along flow lines 5 km downstream and 20 km upstream of the 2012 terminus have high correlation with R-squared values of 0.80 for UI-1 and 0.95 for UI-3.
ISSM$_{PT}$ simulates the major thinning trends as described by Kjær et al. (2012) and Khan et al. (2013) between 1985 and 2010 on UI-1 and UI-3, though not on UI-2. Note that the observed thinning south of UI-3 between 1985 and 1991 (Khan et al., 2013) is not reproduced in ISSM$_{PT}$.

## 4.4  Ice surface velocity

By the end of the ISSM$_{PT}$ simulation, ice flow velocity has doubled at UI-1 and UI-2 and increased by 55 % at UI-3 compared to 1849 (relating the 1849 and 2012 velocity along each flow line between 1 and 30 km upstream the 2012 terminus position). The simulated ice surface velocity evolution in plane view over the study period can be viewed in movie02 (supplementary). Short-term accelerations coincide with the induced ice mass change due to the prescribed terminus change (see movie01, supplementary). The simulation reproduces seasonal and annual velocity variations due to the SMB forcing in the model. Small
(20 m y$^{-1}$) annual velocity fluctuations are forced by seasonal SMB fluctuations. Each retreat from the prescribed terminus change is followed by acceleration between 1 and 70 % and 5–30 % surface lowering, lasting 0.5 to 6 months.

Simulated 2009 ice surface velocity is within ±20 % of observations from Rignot and Mouginot (2012), except in the shear margins, where simulated velocities are up to 250 % higher than observations. Winter velocity maps between 1991 and 2010 (Table 1) are used to evaluate recent changes in simulated velocity. Observed and simulated winter ice surface velocity averaged
between 0 and 5 km and 5 to 10 km upstream of 2012 terminus (Fig. 5) have R-squared values of 0.90 on UI-1, 0.88 on UI-2 and 0.92 on UI-3. Observations show 20 % velocity increase on UI-1 from 07/08 to 08/09, however, this is not captured in ISSM$_{PT}$.

## 5  Discussion

The comparison of ISSM$_{PT}$ and ISSM$_{control}$ shows that the ice surface velocity and thickness are significantly affected by the
prescribed marginal changes. After each prescribed terminus change, ISSM$_{PT}$ simulates short (0.5 to 6 months) periods of faster flow (1–70 % acceleration), and the surface elevation lowers up to 30 % at the new terminus in response to the ice flow acceleration. These are dynamic readjustments to the instantly reduced terminal flow resistance from the prescribed retreat,

which is induced in discrete time steps.

While ISSM$_{PT}$ produces maximum velocity and surface elevation changes of 275 % and 84 % respectively over the simulation period, ISSM$_{control}$ simulates minor changes (maximum $\pm 7$ %) in ice thickness and velocity, representing sole mass changes prescribed by $\Delta$SMB. This highlights the importance of simulated terminus retreat in order to reproduce a UI glacial system
evolution.

In 1985–2002/2005, ISSM$_{PT}$ simulates mass changes similar to observations (Kjær et al., 2012; Khan et al., 2013; Larsen et al., 2016). However, observed and simulated mass changes between 2000 and 2008 differ from each other showing up to 50% more simulated than observed total mass loss (Table 3). During this time, the largest area changes are prescribed at UI-1 with increasing retreat rates from 200–500 m y$^{-1}$ between 1985 and 2005 to 4 km y$^{-1}$ in 2006–2009 along the UI-1 centre
flow line. Between 2006 and 2007, UI-1 splits into three calving fronts, followed by the disintegration of UI-1's floating ice tongue in 2008 (Larsen et al., 2016). The simulated UI-1 terminus is however grounded between 1990 and 2012, only starting to float above an overdeepening in the bathymetry in 2007 in order to be grounded again after the prescribed retreat in 2008. The misfit between observed and simulated mass change could be justified by the absent floating tongue in the simulation. The UI-1 bathymetry is deeper than 500 m below sea level. Therefore, prescribed retreat leads to higher ice mass loss retreating over
the same distance than an observed disintegration of a floating tongue. Moreover, a floating tongue has potential to stabilize the glacier more, decreasing the glacier velocity and dynamical discharge (Nick et al., 2012).

Recent studies suggest dividing mass balance into atmospheric and dynamically driven processes (Nick et al., 2009; Howat and Eddy, 2011; Kjær et al., 2012; Enderlin and Howat, 2013). Our simulation indicates three distinct MB periods when considering $\Delta$SMB and $\Delta$DIL. From the simulation start in 1849 to 1932, the total UI MB is the same for ISSM$_{control}$ and ISSM$_{PT}$, only
diverging five times by one to four Gt y$^{-1}$ when prescribed retreat is enforced. The increasing $\Delta$SMB trend leads to a positive MB and thus mass gain. ISSM$_{PT}$ velocity starts to differ from ISSM$_{control}$ following the first prescribed retreat in 1862, showing a short (< 1 month) acceleration. The simulation indicates stable glacier behaviour without dynamically caused acceleration or thinning.

From 1925 onwards, $\Delta$SMB reveals a negative trend, initiating the negative MB trend that lasts until the simulation end in
2012. Between 1931 and 1992, in two instances (1931–1960 and 1960–1992), 5–7 year periods of sustained less-positive SMB are followed by approximately 20 year long periods of elevated $\Delta$DIL.

Within these 60 years of simulation 31 terminus changes are prescribed, each removing 0.4–5 Gt of ice, which is as much as each of the five terminus changes the preceding 82 years (Fig. 2). The simulated mass loss in this period is therefore highly controlled by the prescribed retreat. $\Delta$DIL consists of the removed ice mass at a prescribed retreat and of changes in ice mass
flux caused by the acceleration of the glacier. We simulate two increased $\Delta$DIL periods preceded by low $\Delta$SMB as the result of observed terminus retreat. Induced by the prescribed terminus change in 1960 and 1966, a new period with increased $\Delta$DIL lasts until 1992.

From 1999 onwards, $\Delta$DIL and $\Delta$SMB are roughly equivalent in contribution to the elevated negative MB. The simulation computes elevated dynamic ice loss due to 5.5 km terminus retreat on UI-1 within 12 years. We can not resolve, whether the
increased retreat of UI-1 is due to (1) the change in $\Delta$SMB from positive to negative values (+7 Gt to −7 Gt) or whether the

glacier itself reaches an unstable position. However, as a result the retreat causes increased simulated $\Delta$DIL adding up to the same amount as the increased negative $\Delta$SMB. UI-2 shows similar behaviour. The result is a negative MB twice as negative as in any year before. In contrast, UI-3 is nearly stable, retreating ~0.5 km between 1999 and 2012 and even advances in some years. It cannot be determined, whether UI-1 and UI-2 also will reach a stable position soon or whether they will continue to

retreat and accelerate.

Although we primarily discuss prescribed ice margin retreat, it is worth mentioning that our method also includes advancing observed terminus position changes at UI-1 and UI-2 in summer 2012 and at UI-3 in the summers 2001, 2003 and 2007. When ice margin advance is prescribed, ice mass is advected downstream and extrapolated over the regions that are newly activated. MEaSUREs data indicate 20 % speed-up on UI-1 from 2007–2008 to 2008–2009, when a large floating ice tongue breaks off

(Larsen et al., 2016). Yet, the observed acceleration is not captured by the simulation and may be related to unresolved loss of buttressing in the simulation.

Simulated frontal changes occur on annual time scales, marked by observation years, and happen thereby less often than in nature. In addition, the magnitude of the removed ice mass is defined by the mesh resolution, set between 300 and 800 m on the area of frontal retreat. Hence, simulated frontal changes appear more abrupt than in nature and do rather capture changes

in glacier velocity and thickness on decadal time scales than a seasonal pattern. The simulation is likely to overestimate the velocity and thickness changes in response to the larger decrease in frontal backstress.

Moreover, the timing of the prescribed terminus changes of ISSM$_{PT}$ is not well constrained, given by observations with gaps of up to 13 years after 1900. In ISSM$_{PT}$, we include 20 additional calving fronts, to minimize the ice mass removed at each terminus change. A simulation excluding the 20 interpolated terminus positions does not affect the overall simulation results

(see supplementary). The total ice mass change shows higher peaks, but results in similar cumulative mass changes. Simulated 2012 ice velocity and thickness of ISSM$_{PT}$ and the sensitivity simulation agree as well with $\pm 5\,\mathrm{m\,yr^{-1}}$ and $\pm 3\,\mathrm{m}$ respectively. However, given more frequent observations could minimize the timing uncertainties and with multiple observed terminus front positions throughout a year, this approach could capture seasonal glacier changes.

## 6  Conclusions

Our study shows that prescribing glacier front positions and surface mass balance are necessary to realistically simulate the multi-decadal evolution of ice velocity and thickness at Upernavik Isstrøm. Our simulation suggests that dynamic response caused by prescribed terminus position change is responsible for 70 % of the total simulated mass change. Thus, moving terminus positions play an important role for UI's acceleration and thinning. The simulation with prescribed terminus changes reproduces distinct mass loss periods of dynamically driven ice mass loss and extends the periods discussed in Kjær et al.

(2012) and Khan et al. (2013) from 1985 to 1932.

Prescribed terminus position change avoids calving and melt rate estimations and reduces simulated retreat uncertainty. Yet, our approach requires knowledge about terminus positions and thus cannot be applied in future projections. However, the simulation results show the importance of calving in order to produce velocity and thickness change of tidewater glaciers. Better

physically based calving laws are needed to understand and predict future glacier behaviour and glacier contribution to sea level rise. With an increasing amount of collected observed terminus front positions, the here discussed method will become a progressively more useful tool for evaluating calving laws or determining calving law parameters for hind-cast simulations before they are applied to future simulations. Short-term simulations with prescribed terminus position changes can determine

what observations are needed to evaluate and construct new calving laws, by establishing if seasonal terminus position variations are necessary to capture long-term glacier behaviour. Future work could include comparisons with simulations using physically based calving laws (e.g. Bondzio et al., 2016; Morlighem et al., 2016) as well as the application of our method to other tidewater glaciers.

*Author contributions.*   KH, NJS, EYL, JEB, KHK, KKK and SHL designed the study and setup the model. KH performed the study and data comparison and led the writing of the manuscript, in which she received input and feedback from all authors. MM created the bed geometry from bathymetry data, supoorted with data from ER and TKD. KKK provided trimline data and observed terminus positions. AMS processed winter ice surface velocity maps from ESA, CCI Greenland.

*Competing interests.*   The authors declare that they have no conflict of interest.

*Acknowledgements.*   The manuscript improved substantially from the editors and reviewers remarks, and the authors would like to thank Olivier Gagliardini, Jeremy Bassis and the two anonymous reviewers for their constructive comments. This study is part of the project "Multi-millennial ice volume changes of the Greenland Ice Sheet" funded by the Geocenter Danmark. KKK was supported by the Danish Council for Independent Research (DFF-4090-00151). We thank Brian Vinter and his team at Niels Bohr Institute, Copenhagen University, for generously supplying high performance computing resources. We wish to thank Camilla Snowman Andresen, Geological Survey of

Greenland and Denmark, for providing bathymetry measurements. Observed termini between 1999 and 2012 were digitized by Trine S. Jensen and Karina Hansen, Geological Survey of Greenland and Denmark. We acknowledge and thank the Ice Sheet System Model group for producing and making available their model. We also acknowledge the use of the DEMs from GIMP, ArcticDEM and Niels Korsgaard and the velocity data provided by ESA (CCI Greenland) and NASA, all available online.

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

**Table 1.** Data for initialising and evaluating the simulation

| Datum | Source | Description |
| --- | --- | --- |
| Bed topography | Morlighem et al. (2017) | Derived with mass conservation approach, extended with bathymetry measurements |
| Bathymetry measurement | | 2012 NASA project, led by Eric Rignot and Todd Dupont |
| Bathymetry measurement | Fenty et al. (2016); OMG Mission. (2016) | NASA project Oceans Melting Greenland OMG |
| Bathymetry measurement | Andresen et al. (2014) | Ship-based single point echo sounders |
| Trimline points | Kjeldsen et al. (2015) | Little Ice Age maximum extent (Fig. 1) |
| Surface mass balance (SMB) | Box (2013) | Monthly data, covering 1840–2012 |
| 1985 Digital elevation model (DEM) | Korsgaard et al. (2016) | Based on aerial photographs, 25 m resolution |
| 2005 DEM | Howat and Eddy (2011) | Greenland Ice Sheet Mapping Project (GIMP), 30 m resolution |
| 2012 DEM | Noh and Howat (2015) | ArcticDEM, 2–10 m resolution |
| Ice surface velocity | Rignot and Mouginot (2012) | Winter 2008/09 |
| Ice surface velocity | http://esa-icesheets-greenland-cci.org/ (described in Nagler et al. (2017)) | Provided by ESA project Climate Change Initiative (CCI) Greenland Ice Sheet in winters between 1991/92 and 2008/09 |
| Ice surface velocity | Howat (2016) | Provided by MEaSUREs, in the winters 2000/01, 2007/08 and 2009/10 |
| Ice surface elevation | Thomas and Studinger (2010); Krabill (2010, updated 2016) | IceBridge ATM; UI-1 in 2009–2012 and UI-3 in 1994, 1999, 2002, 2009, 2010, 2012 |
| Mass change | Wiese et al. (2015); Watkins et al. (2015) | Provided by the Jet Propulsion Laboratory (version: JPL RL05M GRACE mascon solution); suitable for regional (300 km scale) ice sheet mass change comparisons (Schlegel et al., 2016) |

**Table 2.** Steps for model initialisation

| Step | Input | Output |
|---|---|---|
| Relaxation 1 | GIMP extended to 1849 terminus position, Ice viscosity (initial guess), Basal friction (initial guess) | Reconstructed 1849 ice thickness and velocity |
| Thermal | Ice thickness and velocity from relaxation 1 | Improved ice viscosity |
| Inversion | Surface velocity from relaxation 1, Ice viscosity from thermal | Inverted basal friction |
| Relaxation 2 | Ice thickness from relaxation 1, Ice viscosity from thermal, Basal friction from inversion | Steady state ice thickness and velocity |

**Table 3.** Observed vs. simulated ice mass changes (with ISSM$_{PT}$).

| | Khan et al. (2013)[a] | | Larsen et al. (2016) | | |
|---|---|---|---|---|---|
| | $1985 - 2002/05$ | $2002/05 - 2010$ | $2000 - 2005$ | $2006 - 2008$ | $2009 - 2011$ |
| Total observed ice mass changes, Gt | $-32 \pm 9$ | $-17 \pm 10$ | $-6 \pm 20$ | $-25 \pm 14$ | $-39 \pm 17$ |
| Total simulated ice mass changes, Gt | $-37$ [b] | $-32$ [b] | $-48$ | $-41$ | $-44$ |
| Observed dynamic ice loss, Gt | $29 \pm 9$ (90%)[c] | $12 \pm 11$ (80%)[c] | $5 \pm 10$ (80%)[c] | $16 \pm 4$ (62%)[c] | $27 \pm 4$ (68%)[c] |
| Simulated dynamic ice loss, Gt | $32$[b] (86%)[d] | $26$[b] (81%)[d] | $40$ (83%)[d] | $24$ (59%)[d] | $28$ (64%)[d] |

[a] converted from km$^3$ to Gt ice equivalent

[b] results from 2002/05 as mean values of that time

[c] average percentage of total mass change induced by dynamic ice loss

[d] percentage of total mass change that is induced by dynamic ice loss

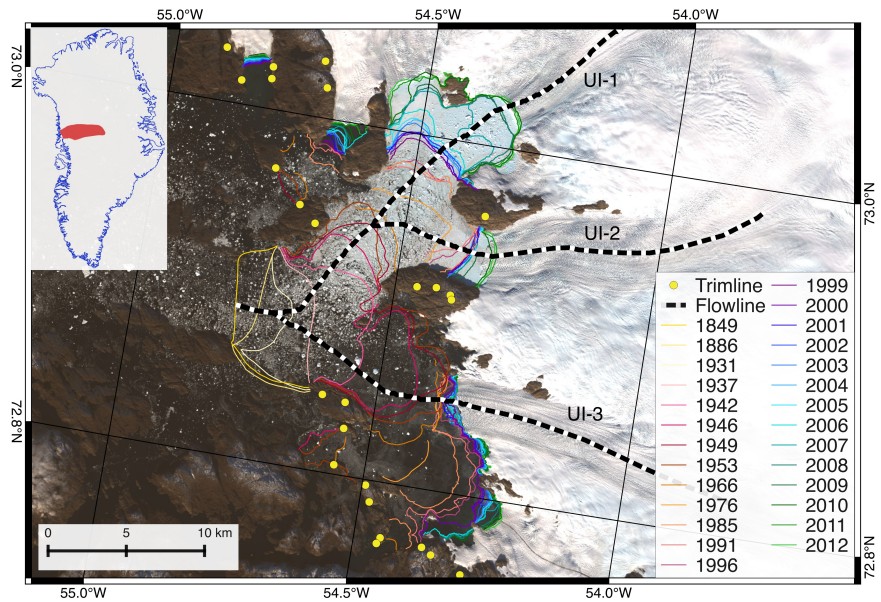

**Figure 1.** Upernavik Isstrøm's observed margin front positions between 1849 and 2012 (lines) and trimline positions (yellow dots; Kjeldsen et al., 2015). The background image is from Landsat 8 (September 2013). Inset is the location and shape of the Upernavik catchment (red area), determined by 2008/09 surface velocity from Rignot and Mouginot (2012), which define the model domain.

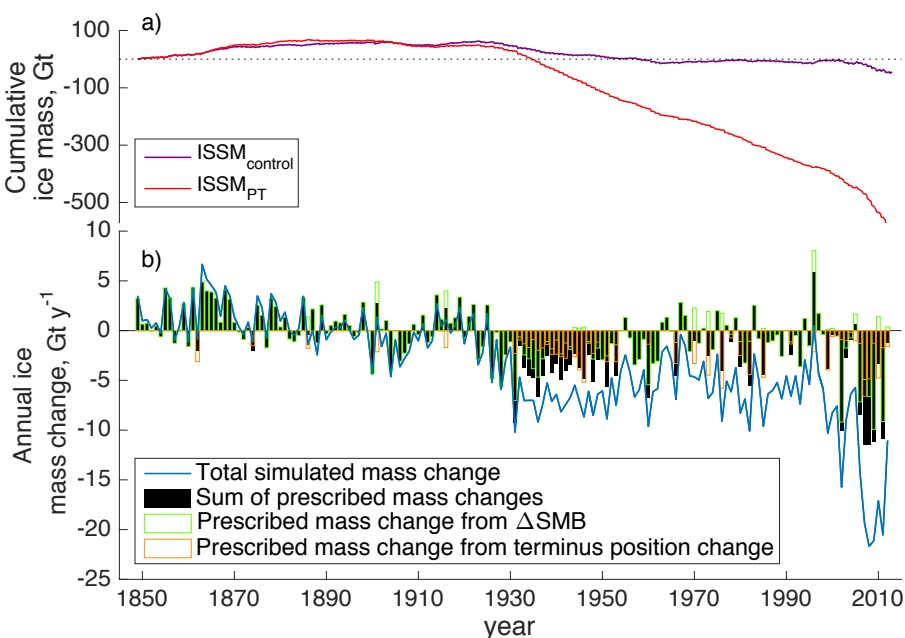

**Figure 2.** a) Simulated cumulative ice mass in Gt. $ISSM_{PT}$ changes are shown in red; control run changes in purple. b) The blue curve illustrates simulated annual change in ice mass for $ISSM_{PT}$. The black bars indicate the ice mass that is removed due to $\Delta SMB$ and prescribed changes of the terminus position. The area between the blue line and black bars corresponds to ice mass changes caused by changes in the ice dynamics that where not prescribed. The green outline marks the portion of mass change due to $\Delta SMB$, and the orange outline the share of prescribed terminus change respectively.

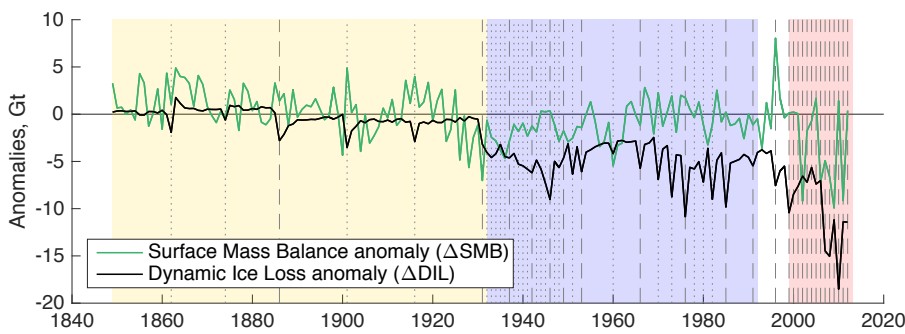

**Figure 3.** Simulated ice mass changes from anomalies (relative to 1964–1990 mean values) for the simulation ISSM$_{PT}$. The background is highlighted in yellow for periods of time where ΔSMB controlled MB, blue is where ice mass loss is driven by ΔDIL and red, where ΔSMB and ΔDIL have equally increased influence on the MB. Prescribed termini changes are marked with dashed (observations) and dotted (interpolations) lines.

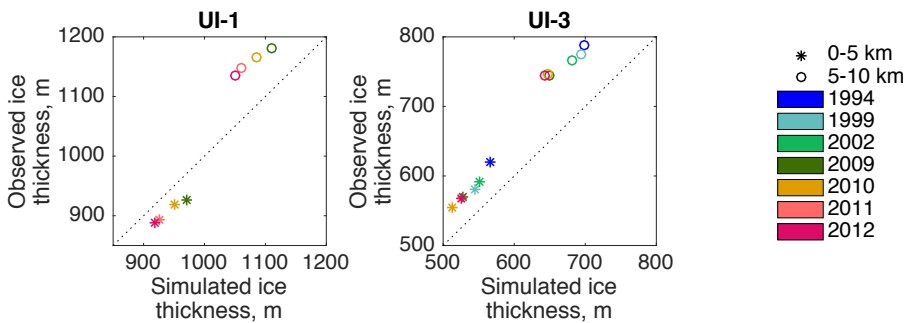

**Figure 4.** Observed vs. simulated ice thickness along flight lines (IceBridge surface elevation data; Thomas and Studinger, 2010; Krabill, 2010, updated 2016) over UI-1 and UI-3. Stars mark mean values between 0 and 5 km from the 2012 terminus, dots refer to mean values 5–10 km upstream from the 2012 terminus. Flight lines over UI-1 are available for the years 2009, 2010, 2011, 2012 and over UI-3 in the years 1994, 1999, 2002, 2009, 2010, 2012.

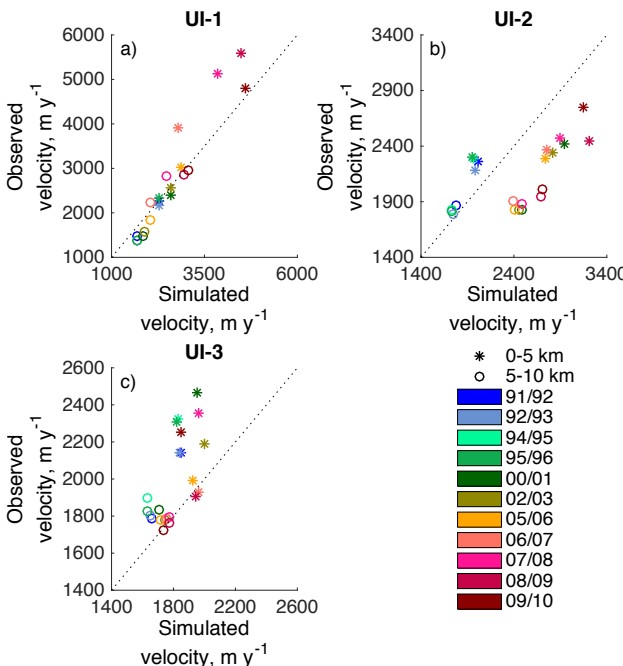

**Figure 5.** Observed vs. simulated ice surface velocity along the central flow lines of UI-1, UI-2 and UI-3. Stars mark mean velocity between 0 and 5 km from the 2012 terminus, dots refer to mean values 5–10 km upstream. Winter velocity maps for 1991/92, 1992/93, 1994/95, 2002/03, 2005/06 and 2008/09 are produced from data available from http://esa-icesheets-greenland-cci.org/ and described in Nagler et al. (2017). Winter velocity maps from 2000/01, 2007/08 and 2009/10 are given by MEaSUREs (Howat, 2016).