# Peer review of "Simulating ice thickness and velocity evolution of Upernavik Isstrøm 1849–2012 by forcing prescribed terminus positions in ISSM"

_The Cryosphere, 2017_

## Referee Comment (RC1) · Anonymous Referee #1 · 9 Aug 2017

General Comments

This study simulates the dynamic response of the 3 main Upernavik Isstrom glaciers to prescribed changes in terminus position and surface mass balance, both from observations, over the period 1849-2012. The authors model ice dynamics using SSA in ISSM and make use of inverse methods to initialise basal conditions. The results of the simulation indicate the importance of terminus position change in driving dynamic change in this glacier system. Overall this paper presents interesting results, but I have significant concerns with regards to the interpretation of the results and the presentation of the methods which lead me to believe it is not yet ready for publication.

This study's approach allows the authors to investigate dynamic response to calving, while circumventing the issue of calving law uncertainty. However, the nature of this approach is such that a large amount of the mass balance change comes from prescribed model inputs. This is not a problem in itself, but the authors have not untangled this model input from model output (i.e. dynamic response) when presenting comparisons with observations. For example, in Section 4.2 and in the supplementary material, the authors show comparisons between modelled and observed mass loss. However, much of this mass loss is actually prescribed through terminus retreat and surface mass balance. It should be trivial to subtract these prescribed components from both the simulated and observed MB. Without this correction the comparison is somewhat misleading, and makes it impossible to assess the performance of the model.

My most serious concern with this manuscript is the claim made in the abstract and in the text that the model matches observations within 20%. In the abstract, the claim applies to the surface elevation and velocity over the period 1990-2012, while in the conclusions, the authors seem to claim that the entire 164 year simulation matches observations within 20%. From the data provided in Figures 4 and 5, and in Sections 4.3 and 4.4, neither claim appears to be accurate. This might be simply fixed by qualifying the statements somewhat, but it leads me to question the accuracy of the other (currently unverifiable) claims about the match between model and observation. I would like to see additional figures showing the mismatch in elevation and velocity across the domain to back up the claims made in Sections 4.3 and 4.4.

The description of the model setup, physics, boundary conditions and initialisation is somewhat unclear and significantly lacking in detail. What does the model domain look like? Is it defined by the ice catchment? Does it extend to the ice divide? What velocity data are used to invert for basal friction? What happens to the basal friction condition when flotation is achieved?

In comparing surface elevations, the authors state that the surface lies within 20% of observations, and similar percentage comparisons of surface elevation are made

throughout the results section (e.g. 84% surface lowering at UI-2 2012 terminus). This should be restated in terms of ice thickness, which is altitude-agnostic and which, after all, is the variable of interest from an ice dynamics perspective. A 20% error/change in surface elevation translates into quite different thickness errors/changes depending on whether the ice is floating or resting on bedrock at 500 m.a.s.l.

I found quite a few grammar/language errors, some of which I have highlighted in 'technical corrections' below.

Specific Comments

P1 L3: make it clearer that you prescribe changing terminus position. "Observed glacier terminus changes" could be e.g. oceanic or atmospheric conditions.

P1 L5: I think you used 2012 velocities to invert for basal drag (though I'm not sure), and terminus positions (and SMB) are prescribed. As such, I don't think a <20% error in elevation and velocity at the end of your simulation would necessarily imply that your model is realistic from 1849-2012. It would tell you that your basal inversion worked properly. But more importantly, this is not accurate! For example, Fig 4 shows UI-1 observed surface elevation in 2009 at 5-10km of over 500 m.a.s.l., but modelled is less than 400 m.a.s.l. Fig 5 shows UI-2 0-5km 08/09 observed velocity is just under 2500m/a, but modelled is over 3000 m/a. You explicitly state in the text that simulated 2012 upstream surface elevation is 56-62% of that observed. And these are data averaged over a large area. In the shear margins, you mismatch by 100%. This in itself is not a problem – shear margins are tricky, but you cannot claim that you match elevation and velocity within 20%.

P2 L12: What is a "dynamic ice loss event"? In the context of Kjaer et al (2012), it seems to be a multi-year period of sustained accelerated calving. You should clarify this.

P2 L14: The final sentence of this paragraph feels out of place. Perhaps move it to the

start of the next paragraph. "Hence" here implies that the focus of previous studies is a result of the two dynamic mass loss events.

P3 Fig 1: This is a good figure, but the poor contrast in the landsat image between rock and ocean makes it slightly tricky to pick out the historic positions of the individual glaciers. Perhaps you could tweak the bands a little?

P3 L4,5: This sentence is quite unclear. It starts by describing SSA (approximation for stokes, long. stress), but "neglecting lateral drag" is not a fundamental part of SSA. I guess you mean that you choose to neglect lateral drag on the sides of your domain? Given the width of the domain, this is quite justifiable, but explain it better and give this justification. I also think you could give a more technical and less clunky description of longitudinal stress gradients.

P3 L10: Can you show, or at least properly describe, the domain somewhere?

P3 L6: Why use surface air temperature for depth integrated viscosity? Is there any reason to think that surface air temp is equal to, or even correlates with, internal temperature?

P4 L8: Can you give more details on the extrapolation of velocity? Is this done using a mass conservation approach? I assume that is what is meant by "following fjord bathymetry"? If so, were changes in glacier width also accounted for?

P5 L7: It took me some time to figure out your strategy here, but now I see that your interpolated surface elevation and bathymetry tells you whether the ice should be floating or grounded, and therefore gives you a thickness. Maybe you could clarify this?

P5 L19: What about floating regions? I guess driving stress is small (but non-zero) here.

P5 L23: Authors state "the first relaxation... provides ice thickness and velocity for the second relaxation. Given computed ice velocity from the first relaxation, basal friction can be redefined". So, is the inversion done with respect to observed velocity

or simulated velocity from the previous relaxation? I guess the former, in which case you should clarify the above statements; given the instantaneousness of the stokes equations, I don't think the velocity from the first relaxation really feeds into the second relaxation at all, except perhaps to provide the initial guess for viscosity in your first iteration. If the latter, this feels questionable – using velocity from SIA basal drag in SSA model to invert for new basal drag...

P6 L9: If you want to show relative changes, you should be looking at thickness, as mentioned above.

P7 L1: How much of this -585 Gt was prescribed?

P7 L8: "hereafter anomalies deltaSMB and deltaDIL" - I see what you mean, but this isn't a sentence.

P7 L16-21: This paragraph and associated table are not very intuitive and could be improved. "2002/05 – 2010" should be clarified in the text – it's not clear what this range represents. The authors state that mass balance corresponds to three sets of cited observations, but only two are present in the table. It's also somewhat confusing that you mix comparisons of observed and modelled mass balance with comparisons of DIL % - this is made even more confusing by the lack of these % DIL values in the table. I'd recommend adding some data on the % DIL and SMB from simulation and observations to Table 2. This would significantly clarify the last sentence, in which the authors state that % DIL agrees with Khan and Larsen – the reader is drawn to Table 2 for evidence of this agreement, but none is provided. Also, as mentioned in general comments, you need to untangle the prescribed and resultant mass loss before comparing with observations.

P8 Table 2: I guess the simulated changes don't appear to correspond because Khan 2013 don't measure changes in the whole domain of your model? It would be worth explaining this, otherwise readers might wonder how the 2002/05-2010 simulated mass loss is 32 Gt, but the 2000-2011 mass loss totals 133 Gt.

P8 L4: As mentioned in general comments, I think you should discuss thickness changes, or else stick to absolute values.

P10 L9: Source for these winter velocity maps?

P10 L23: "ice surface elevation... velocity observations". This doesn't seem to make sense.

P11 L15: "The simulation reproduces not only the retreat..." - I don't think you can say that the model reproduces the observed retreat and advance. You prescribe these changes.

P11 L27: "matching observed velocity, surface elevation and mass changes within 20% of observations". As stated in general comments above, I think your comparisons with observations are flawed at present. Furthermore, Table 2, Fig. 4, Fig. 5 demonstrate that this figure of 20% is not accurate.

Technical Corrections

P1 L6: "and are within"

P1 L7: "Increased ice flow acceleration", surely its just "ice flow acceleration" or "increased ice velocity"?

P3 L10: "The grounding line", no need to capitalize.

P5 L21: "The basal friction", ditto.

P6 L7: "away form"

P6 L2: "that are causing numerical instabilities" is not good english here.

P7 Fig 3: Caption refers to SID rather than DIL.

---

## Referee Comment (RC2) · J. Bassis (Referee) · 14 Sep 2017

The goal of this study is to simulate the dynamic response of the Upervnavik Isstrom glacier system from 1849-2012 to prescribed terminus changes combined with changes in surface mass balance. The authors use the ISSM model approximation to the well known shallow shelf approximation (SSA) equations. The authors find that prescribed changes in terminus position have a large effect on dynamic discharge, reinforcing many prior studies that came to similar findings. Overall, one of the strengths of this study is that it is able to simulate dynamic drawdown over a relatively long period of time. I did, however, have some questions.

[Figure]

1. I am a bit confused by the geometry of the glacier system. It says in several places that the grounding line is computed automatically by the model. I guess this means that the groundling line (transition from grounded ice to floating ice) is computed automatically by the model, but the terminus position or calving front position is floating and this position is specified? The existence of an ice tongue or shelf is not obvious at all from the discussion or figures: does the glacier always have a floating tongue or is the floating tongue only there part of the time. After reading through a few more times, I started to think that there is no ice tongue and the terminus is grounded, consistent with this system being a tidewater glacier system, but then there isn't a grounding line. Overall, I would like the authors to be a bit more careful with their explanation of whether the glacier has an actual grounding line and if the grounding line evolves separate from the calving front or whether there is really a calving front, which may sometimes approach floatation or something else.

2. The model description could use a bit more detail. As far as I can tell the authors are using the SSA approximation as implemented in ISSM and inverting for basal friction to best match observed velocities. This is acceptable, but the authors should also tell us which sliding law was used. Back in the old days, models used to use a sliding law with friction proportional to velocity (Newtonian) because it was easy to implement. Now we know that the sliding law is rarely Newtonian, but some prefer a plastic bed, Coulomb plastic, Weertman or some combination of the three. For reasons related to my next point, it is important to provide the sliding law. The authors should also probably provide a map of the inferred basal friction parameter.

3. As I understand it, the authors apply a prescribed surface mass balance along with prescribed changes in terminus position to simulate changes to the dynamics of the glacier system. Moreover, the authors use an inverse method to invert for the basal friction coefficients in their sliding law. Now ice dynamics models are based on approximations to the Stokes equations. A consequence of this is that if the geometry is appropriately specified and the boundary conditions are all correctly specified the

velocity is completely determined. Because the authors are using observed velocities to tune the friction coefficients and prescribing changes in terminus position, it isn't all that surprising that they can simulate the correct dynamic response. In fact, it would be surprising if the model failed given the tuning step. What is surprising and impressive is that authors are able to get the correct dynamic response over a fairly long time interval. This seems like it is probably dictated, at least in part, by the choice of sliding law—which is why I think it is important to specify the sliding law and show us its pattern of spatial variation. Also, are the model results sensitive to the form or magnitude of the sliding coefficient. For example, do you get similar results for plastic, Coulomb and Weertman type sliding laws? How sensitive are the results to the inversion? Is the good agreement a consequence of extensive model tuning or relatively insensitive to model tuning? Related to this, the authors need to be careful when comparing observations of velocity with simulated velocity. Good agreement means the inversion was able to match surface velocities, but tells us nothing about the models skill.

4. Related to this, I'm a bit confused by the metrics for model success. It seems to me as though the authors are comparing observations of mass balance to simulations of mass balance. However, surface mass balance is prescribed and changes in terminus position are prescribed so the only part of this that can vary is the increased dynamic discharge. Why not just compare simulated change in dynamic discharge predicted by their model with that inferred from observations. For example, Figure 2 shows annual mass loss along with change in mass loss from prescribed changes in terminus position. What about also showing mass loss from prescribed surface mass balance? Then we would clearly see the component that is predicted (dynamic discharge) and what is specified. The more I think about it, it seems as though the observations probably give changes in trim line (or something like this) so maybe the right comparison is between predicted glacier surface elevation and observed trim lines (as opposed to ice thickness)? (I apologize to the authors if I misunderstood their data or comparisons). As I said before, I also don't think that the authors can claim that the match between measured velocities and observed velocities provides any test of the model.

Friction has been determined by tuning the model to match observed velocities so any match between observed and simulated velocities is partly a consequence of the tuning procedure. Here I think the authors might be able to narrate to readers a bit more thoroughly what they actually measured and what they predicted (without prescribing) and how the measurements can be used to test the things that the model predicted that weren't ingested in any tuning exercises.

Some miscellaneous comments: Page line 13 extra space before

Don't capitalize Grounding line or Basal friction

Page 3 below 5: The ice temperature is determine by solving an advection-diffusion equation. The paper says the temperature field was initialized using surface air temperature. Does this mean the ice temperature was run to steady-state using the assumed surface air temperature?

Page 3, line 10: The grounding line position is automatically calculated in each step implies that the terminus is at flotation. Why is this a good approximation and why is it forbidden for the terminus to have a thickness greater than flotation?

I found a few other grammar mistakes throughout and I would urge the authors to give the manuscript one more proof read.

---

## Author Comment (AC1) · 12 Oct 2017

**Interactive response to reviewer comments on* "Simulating ice thickness and velocity evolution of Upernavik Isstrøm 1849–2012 by forcing prescribed terminus positions in ISSM"**

**by Konstanze Haubner and co-authors**

We thank both reviewers for their constructive comments on our manuscript. We feel the requested changes have improved the clarity of the paper and appreciate their feedback. The author response to reviewers is structured as follows:

– Reviewers' comments in blue

– Authors' response in black

5  We significantly rewrote some sections of the manuscript, to improve clarity. The major changes include:

– Model evaluation includes now comparisons of observed and simulated ice thickness instead of surface elevation

– Additional figures in the supplementary showing the basal friction coefficient and spatial comparisons for simulated and observed ice thickness, velocity

– New table visualising model initialisation steps

10  – Including the co-authors Eric Rignot and Todd K. Dupont (TC editorial support (Svenja Lange) is informed)

Our conclusions remain unchanged. The reviewed manuscript with tracked changes is attached.

**Referee #1 (anonymous)**

**General comments**

This study simulates the dynamic response of the 3 main Upernavik Isstrom glaciers to prescribed changes in terminus position
15  and surface mass balance, both from observations, over the period 1849–2012. The authors model ice dynamics using SSA in
ISSM and make use of inverse methods to initialise basal conditions. The results of the simulation indicate the importance of
terminus position change in driving dynamic change in this glacier system. Overall this paper presents interesting results, but I
have significant concerns with regards to the interpretation of the results and the presentation of the methods which lead me to
believe it is not yet ready for publication.

20  This study's approach allows the authors to investigate dynamic response to calving, while circumventing the issue of calving
law uncertainty. However, the nature of this approach is such that a large amount of the mass balance change comes from
prescribed model inputs. This is not a problem in itself, but the authors have not untangled this model input from model output

(i.e. dynamic response) when presenting comparisons with observations. For example, in Section 4.2 and in the supplementary material, the authors show comparisons between modelled and observed mass loss. However, much of this mass loss is actually prescribed through terminus retreat and surface mass balance. It should be trivial to subtract these prescribed components from both the simulated and observed MB. Without this correction the comparison is somewhat misleading, and makes it impossible to assess the performance of the model.

In our manuscript, we divide the simulated mass changes into mass change due to prescribed SMB and dynamic component DIL. We agree that a division of $\Delta$DIL into mass change prescribed by terminus position change and the simulated dynamical mass change improves assessing model performance. Figure 2b is adapted accordingly. We gain knowledge about the contributions to mass loss from prescribed SMB, terminus change and resulting thinning and acceleration: "[...] while 30 % of total ice mass loss simulated by ISSM$_{PT}$ was prescribed, with $\Delta$SMB accounting for 9 % ($-50$ Gt) and prescribed terminus position change contributed 21 % ($-121$ Gt). Thus, 70 % of by ISSM$_{PT}$ simulated mass loss is caused by thinning and acceleration."
However, the choice of dividing the mass loss into $\Delta$SMB and $\Delta$DIL is motivated by the cited publications Khan et al. (2013) and Larsen et al. (2016). They compute mass change by comparing Upernavik's ice surface elevation in different periods, compute $\Delta$SMB between the dates of the surface elevations and compute $\Delta$DIL as residual of computed mass change and SMB. By comparing our simulated $\Delta$DIL with those observations, we find it important to use the same measure and compute prescribed $\Delta$SMB and simulated mass changes on the domains observed in Khan et al. (2013) and Larsen et al. (2016). We also use the same term: dynamic ice loss.
We include explanations about the background of the mass comparison with Khan et al. (2013) and Larsen et al. (2016) and re-write the subsection 4.2.

My most serious concern with this manuscript is the claim made in the abstract and in the text that the model matches observations within 20%. In the abstract, the claim applies to the surface elevation and velocity over the period 1990–2012, while in the conclusions, the authors seem to claim that the entire 164 year simulation matches observations within 20%. From the data provided in Figures 4 and 5, and in Sections 4.3 and 4.4, neither claim appears to be accurate. This might be simply fixed by qualifying the statements somewhat, but it leads me to question the accuracy of the other (currently unverifiable) claims about the match between model and observation. I would like to see additional figures showing the mismatch in elevation and velocity across the domain to back up the claims made in Sections 4.3 and 4.4.

We remove claims similar to simulation matching $\pm 20\%$ observations and change all evaluations of simulated ice surface elevation to ice thickness. We add figures to the supplementary, showing spatial absolute and relative differenced between simulated and observed ice thickness.
Comparing simulated and observed ice thickness changes the percentage of accuracy. We adapt the text accordingly.

The description of the model setup, physics, boundary conditions and initialisation is somewhat unclear and significantly lacking in detail. What does the model domain look like? Is it defined by the ice catchment? Does it extend to the ice divide?

The model domain is defined by the ice catchment and extends to the ice domain, marked as a red area Figure 1. The following sentence is now included in the model description: "The model domain is set to the Upernavik catchment, which is defined by the flow direction given by the 2008/09 surface velocity from Rignot and Mouginot (2012) (red area in Fig. **??**)."

What velocity data are used to invert for basal friction? What happens to the basal friction condition when flotation is achieved?

Velocity gained from relaxation is used to invert for the basal friction coefficient. Now, Table 2 provides more information about the model initialization steps. The section describing model initialisation is re-written to improve clarity.

Friction is not applied on floating areas.

5   In comparing surface elevations, the authors state that the surface lies within 20% of observations, and similar percentage comparisons of surface elevation are made throughout the results section (e.g. 84% surface lowering at UI-2 2012 terminus). This should be restated in terms of ice thickness, which is altitude-agnostic and which, after all, is the variable of interest from an ice dynamics perspective. A 20% error/change in surface elevation translates into quite different thickness errors/changes depending on whether the ice is floating or resting on bedrock at 500 m.a.s.l.

10  As explained above, we change surface elevation comparisons to thickness comparisons and adapt all numbers accordingly.

I found quite a few grammar/language errors, some of which I have highlighted in 'technical corrections' below.

We improved the study regarding grammar and spelling.

**Specific Comments**

15  P1 L3: make it clearer that you prescribe changing terminus position. "Observed glacier terminus changes" could be e.g. oceanic or atmospheric conditions.

Sentence has changed during revision.

P1 L5: I think you used 2012 velocities to invert for basal drag (though I'm not sure), and terminus positions (and SMB) are prescribed. As such, I don't think a <20% error in elevation and velocity at the end of your simulation would necessarily imply

20  that your model is realistic from 1849-2012. It would tell you that your basal inversion worked properly. But more importantly, this is not accurate! For example, Fig 4 shows UI-1 observed surface elevation in 2009 at 5–10km of over 500 m.a.s.l., but modelled is less than 400 m.a.s.l. Fig 5 shows UI-2 0-5km 08/09 observed velocity is just under 2500m/a, but modelled is over 3000 m/a. You explicitly state in the text that simulated 2012 upstream surface elevation is 56-62% of that observed. And these are data averaged over a large area. In the shear margins, you mismatch by 100%. This in itself is not a problem – shear

25  margins are tricky, but you cannot claim that you match elevation and velocity within 20%.

We agree, the statement is too strong and formulated misleadingly. Conclusions and abstract are revised and focus now on different mass loss periods, the different contributors and how PT could be used to improve ice sheet models.

P2 L12: What is a "dynamic ice loss event"? In the context of Kjaer et al (2012), it seems to be a multi-year period of sustained accelerated calving. You should clarify this.

30  We re-phrase to "periods of increased dynamically driven ice loss".

P2 L14: The final sentence of this paragraph feels out of place. Perhaps move it to the start of the next paragraph. "Hence" here implies that the focus of previous studies is a result of the two dynamic mass loss events.

We moved the sentence to the next paragraph and start the sentence with "Previous studies [...]"

P3 Fig 1: This is a good figure, but the poor contrast in the landsat image between rock and ocean makes it slightly tricky to

35  pick out the historic positions of the individual glaciers. Perhaps you could tweak the bands a little?

We changed the color bands and contrast aiming for better rock-ocean contrast.

P3 L4,5: This sentence is quite unclear. It starts by describing SSA (approximation for stokes, long. stress), but "neglecting lateral drag" is not a fundamental part of SSA. I guess you mean that you choose to neglect lateral drag on the sides of your domain? Given the width of the domain, this is quite justifiable, but explain it better and give this justification. I also think you could give a more technical and less clunky description of longitudinal stress gradients.

We re-write the paragraph to: "Ice flow is calculated applying the Shelfy Stream Approximation (SSA; MacAyeal, 1989), that integrates vertically averaged ice properties (e.g. ice rheology, thickness, velocity) and neglects vertical shear stresses. The SSA is well suited for fast-flowing glaciers like Upernavik, where the ice flow is primarily driven by basal sliding."

P3 L10: Can you show, or at least properly describe, the domain somewhere?

The model domain is the UI catchment and shown in Figure 1. The figure caption now contains information about the model domain and the following sentence is added to the model description:

"The model domain is set to the Upernavik catchment, which is defined by the flow direction given by the 2008/09 surface velocity from Rignot and Mouginot (2012) (red area in Fig. 1)."

P3 L6: Why use surface air temperature for depth integrated viscosity? Is there any reason to think that surface air temp is equal to, or even correlates with, internal temperature?

We divided the explanation for ice viscosity, and moved some parts of the explanation to section 3.1, Model Initialisation. The first part remains in the Introduction section in section 3:

"Ice viscosity follows Glen's Flow law (Glen, 1955). The initial viscosity is taken from Table 3.4 in Cuffey and Paterson (2010, p. 75), assuming ice temperature of $-5°C$ and will be refined in section 3.1"

Further explanation is given in section 3.1 ("Model initialisation"):

"Given computed ice velocity and thickness from the first relaxation, ice viscosity and basal friction can be redefined. The ice viscosity is calculated by extruding the model with 15 layers and solving for the thermal steady state based on forcing the surface with 1854–1900 UI mean surface air temperature (Box, 2013)."

P4 L8: Can you give more details on the extrapolation of velocity? Is this done using a mass conservation approach? I assume that is what is meant by "following fjord bathymetry"? If so, were changes in glacier width also accounted for?

We re-phrased the paragraph to clarify the initialisation steps and added Table 2 to give an overview, which steps are performed including their goals.

The initial velocity is derived from stress balance solution, given GIMP surface elevation extended to the 1849 terminus as described in section 3.1. For the initial ice velocity, we asume driving stress to be equal to basal stress at any given point.

P5 L7: It took me some time to figure out your strategy here, but now I see that your interpolated surface elevation and bathymetry tells you whether the ice should be floating or grounded, and therefore gives you a thickness. Maybe you could clarify this?

We reduce the explanation to "The ice thickness is set to floatation height or to the maximum thickness, defined through the initialised ice surface elevation and bed topography. "

P5 L19: What about floating regions? I guess driving stress is small (but non-zero) here.

Friction is not applied on floating regions. The driving stress is small.

P5 L23: Authors state "the first relaxation... provides ice thickness and velocity for the second relaxation. Given computed ice velocity from the first relaxation, basal friction can be redefined". So, is the inversion done with respect to observed velocity or simulated velocity from the previous relaxation? I guess the former, in which case you should clarify the above statements; given the instantaneousness of the stokes equations, I don't think the velocity from the first relaxation really feeds into the second relaxation at all, except perhaps to provide the initial guess for viscosity in your first iteration. If the latter, this feels questionable – using velocity from SIA basal drag in SSA model to invert for new basal drag...

We invert for previously relaxed surface velocity after improving the ice viscosity by obtaining a thermal steady state.

New inversion description: "Given computed ice velocity and thickness from the first relaxation, ice viscosity and basal friction can be redefined. The ice viscosity is calculated by extruding the model with 15 layers and solving for the thermal steady state based on forcing the surface with 1854–1900 UI mean surface air temperature (Box, 2013). The basal friction coefficient is constant in time, but varies in space, and is calculated by an adjoint-based inversion, following Morlighem et al. (2010) and MacAyeal (1993), given the updated ice viscosity from the thermal steady state simulation."

P6 L9: If you want to show relative changes, you should be looking at thickness, as mentioned above.

Done. (see answer above)

We added "99 % of simulated ISSM$_{control}$ mass loss was prescribed by $\Delta$SMB while 30 % of total ice mass loss simulated by ISSM$_{PT}$ was prescribed, with $\Delta$SMB accounting for 9 % ($-50$ Gt) and prescribed terminus position change contributed 21 % ($-121$ Gt). Thus, 70 % of by ISSM$_{PT}$ simulated mass loss is caused by thinning and acceleration."

P7 L8: "hereafter anomalies deltaSMB and deltaDIL" - I see what you mean, but this isn't a sentence.

Now: "hereafter referred to as anomalies $\Delta$SMB and $\Delta$DIL"

P7 L16-21: This paragraph and associated table are not very intuitive and could be improved. "2002/05 – 2010" should be clarified in the text – it's not clear what this range represents. The authors state that mass balance corresponds to three sets of cited observations, but only two are present in the table. It's also somewhat confusing that you mix comparisons of observed and modelled mass balance with comparisons of DIL % - this is made even more confusing by the lack of these % DIL values in the table. I'd recommend adding some data on the % DIL and SMB from simulation and observations to Table 2. This would significantly clarify the last sentence, in which the authors state that % DIL agrees with Khan and Larsen – the reader is drawn to Table 2 for evidence of this agreement, but none is provided. Also, as mentioned in general comments, you need to untangle the prescribed and resultant mass loss before comparing with observations.

P8 Table 2: I guess the simulated changes don't appear to correspond because Khan 2013 don't measure changes in the whole domain of your model? It would be worth explaining this, otherwise readers might wonder how the 2002/05-2010 simulated mass loss is 32 Gt, but the 2000–2011 mass loss totals 133 Gt.

We added a paragraph and extended Table 3 to address both comments above:

"Khan et al. (2013) and Larsen et al. (2016) measure surface elevation changes from aerial photographs, satellites and digital elevation models between 1985 and 2010. These yield a total mass change during different time periods and congruent to our

calculations ΔDIL is estimated as the residual of mass change and ΔSMB. Both studies refer to different areas within the UI catchment. Table 3 presents a comparison of the observed mass changes and our simulation results, recalculated for the particular areas. Due to sparse data coverage Khan et al. (2013) combine surface elevation measurements acquired between 2002 and 2005 to quantify elevation changes and refer to this period as 2002/2005. The average of simulated ice mass loss

5   between 2002 and 2005 is taken for comparison with the 2002/2005 observations from Khan et al. (2013)."

P8 L4: As mentioned in general comments, I think you should discuss thickness changes, or else stick to absolute values.

Answered above.

P10 L9: Source for these winter velocity maps?

The winter velocity maps are produced from data available from http://esa-icesheets-greenland-cci.org/ and described in Na-

10  gler et al 2017.

We added the information to Table 1 and the caption of Figure 5.

P10 L23: "ice surface elevation. . . velocity observations". This doesn't seem to make sense.

The sentence changed during revision.

P11 L15: "The simulation reproduces not only the retreat..." - I don't think you can say that the model reproduces the observed

15  retreat and advance. You prescribe these changes.

This sentence was changed to clarify its statement: "Although we primarily discuss prescribed ice margin retreat, it is worth mentioning that our method also includes advancing observed terminus position changes at UI-1 and UI-2 in summer 2012 and at UI-3 in the summers 2001, 2003 and 2007."

P11 L27: "matching observed velocity, surface elevation and mass changes within 20% of observations". As stated in general

20  comments above, I think your comparisons with observations are flawed at present. Furthermore, Table 2, Fig. 4, Fig. 5 demon-strate that this figure of 20% is not accurate.

Changed. (See general comments above).

**Technical Corrections**

P1 L6: "and are within"

25  We deleted "and".

P1 L7: "Increased ice flow acceleration", surely its just "ice flow acceleration" or "increased ice velocity"?

Changed to "increased ice velocity".

P3 L10: "The grounding line", no need to capitalize.

Done.

30  P5 L21: "The basal friction", ditto.

Done.

P6 L7: "away form"

Done.

P6 L2: "that are causing numerical instabilities" is not good english here.

35  Changed to "The additional calving fronts aim to improve realistic simulation behavior by splitting large ice area changes

induced by the prescribed terminus changes into smaller areas within shorter time periods."

P7 Fig 3: Caption refers to SID rather than DIL.

Caption was changed.

**References**

Box, J. E.: Greenland ice sheet mass balance reconstruction. Part II: Surface mass balance (1840-2010), Journal of Climate, 26, doi:10.1175/JCLI-D-12-00518.1, 2013.

Cuffey, K. and Paterson, W.: The Physics of Glaciers, Elsevier Science, 2010.

5 Glen, J. W.: The creep of polycrystalline ice, Proceedings of the Royal Society of London A: Mathematical, Physical and Engineering Sciences, 228, 519–538, doi:10.1098/rspa.1955.0066, 1955.

Khan, S. A., KjæR, K. H., Korsgaard, N. J., Wahr, J., Joughin, I. R., Timm, L. H., Bamber, J. L., Broeke, M. R., Stearns, L. A., Hamilton, G. S., Csatho, B. M., Nielsen, K., Hurkmans, R., and Babonis, G.: Recurring dynamically induced thinning during 1985 to 2010 on Upernavik Isstrøm, West Greenland, Journal of Geophysical Research (Earth Surface), 118, 111–121, doi:10.1029/2012JF002481, 2013.

10 Larsen, S. H., Khan, S. A., Ahlstrøm, A. P., Hvidberg, C. S., Willis, M. J., and Andersen, S. B.: Increased mass loss and asynchronous behavior of marine-terminating outlet glaciers at Upernavik Isstrøm, NW Greenland, Journal of Geophysical Research (Earth Surface), 121, 241–256, doi:10.1002/2015JF003507, 2016.

MacAyeal, D. R.: Large-scale ice flow over a viscous basal sediment - Theory and application to ice stream B, Antarctica, Journal of Geophysical Research, 94, 4071–4087, doi:10.1029/JB094iB04p04071, 1989.

15 MacAyeal, D. R.: Binge/purge oscillations of the Laurentide Ice Sheet as a cause of the North Atlantic's Heinrich events, Paleoceanography, 8, 775–784, doi:10.1029/93PA02200, 1993.

Morlighem, M., Rignot, E., Seroussi, H., Larour, E., Ben Dhia, H., and Aubry, D.: Spatial patterns of basal drag inferred using control methods from a full-Stokes and simpler models for Pine Island Glacier, West Antarctica, Geophysical Research Letters, 37, L14 502, doi:10.1029/2010GL043853, 2010.

20 Rignot, E. and Mouginot, J.: Ice flow in Greenland for the International Polar Year 2008-2009, Geophysical Research Letters, 39, L11 501, doi:10.1029/2012GL051634, 2012.

---

## Author Comment (AC2) · 12 Oct 2017

*Interactive response to reviewer comments on* "Simulating ice thickness and velocity evolution of Upernavik Isstrøm 1849–2012 by forcing prescribed terminus positions in ISSM"

 by Konstanze Haubner and co-authors

We thank both reviewers for their constructive comments on our manuscript. We feel the requested changes have improved the clarity of the paper and appreciate their feedback. The author response to reviewers is structured as follows:

– Reviewers' comments in blue

– Authors' response in black

We significantly rewrote some sections of the manuscript, to improve clarity. The major changes include:

– Model evaluation includes now comparisons of observed and simulated ice thickness instead of surface elevation

– Additional figures in the supplementary showing the basal friction coefficient and spatial comparisons for simulated and observed ice thickness, velocity

– New table visualising model initialisation steps

– Including the co-authors Eric Rignot and Todd K. Dupont (TC editorial support (Svenja Lange) is informed)

Our conclusions remain unchanged. The reviewed manuscript with tracked changes is attached.

**Referee #2 (J. Bassis)**

**General comments**

The goal of this study is to simulate the dynamic response of the Upervnavik Isstrom glacier system from 1849-2012 to prescribed terminus changes combined with changes in surface mass balance. The authors use the ISSM model approximation to the well known shallow shelf approximation (SSA) equations. The authors find that prescribed changes in terminus position have a large effect on dynamic discharge, reinforcing many prior studies that came to similar findings. Overall, one of the strengths of this study is that it is able to simulate dynamic drawdown over a relatively long period of time. I did, however, have some questions.

1. I am a bit confused by the geometry of the glacier system. It says in several places that the grounding line is computed automatically by the model. I guess this means that the groundling line (transition from grounded ice to floating ice) is computed automatically by the model, but the terminus position or calving front position is floating and this position is specified? The

existence of an ice tongue or shelf is not obvious at all from the discussion or figures: does the glacier always have a floating tongue or is the floating tongue only there part of the time. After reading through a few more times, I started to think that there is no ice tongue and the terminus is grounded, consistent with this system being a tidewater glacier system, but then there isn't a grounding line. Overall, I would like the authors to be a bit more careful with their explanation of whether the glacier has an actual grounding line and if the grounding line evolves separate from the calving front or whether there is really a calving front, which may sometimes approach floatation or something else.

We agree that the text was not very clear: the ice front position is prescribed, and ice is allowed to float depending on a hydrostatic criterion (Seroussi et al., 2014). Most of the time, the calving front is grounded (see video01, supplementary). Though, UI-3 shows an evolving floating tongue between 1900 and 1951.

In order to improve the understanding of the grounding line integration, we changed the manuscript in the following way: We removed grounding line in the abstract to take away to focus of it being a product of our method. We added the following sentences in section 3 and 3.2:

"The ice is allowed to float depending on a hydrostatic criterion (Seroussi et al., 2014)."

"Within the prescribed ice area, the grounding line is evolving freely and floating tongue formation is thereby allowed."

2. The model description could use a bit more detail. As far as I can tell the authors are using the SSA approximation as implemented in ISSM and inverting for basal friction to best match observed velocities. This is acceptable, but the authors should also tell us which sliding law was used. Back in the old days, models used to use a sliding law with friction proportional to velocity (Newtonian) because it was easy to implement. Now we know that the sliding law is rarely Newtonian, but some prefer a plastic bed, Coulomb plastic, Weertman or some combination of the three. For reasons related to my next point, it is important to provide the sliding law. The authors should also probably provide a map of the inferred basal friction parameter.

To clarify the used sliding law we include the tex below in section 3 and Figure 1 into the supplementary:

A Coulomb-like friction law is applied on grounded ice:

$$\tau_b = -C^2 N v_b \tag{1}$$

where $v_b$ is the basal velocity, $N$ the effective pressure on the glacier base and $C$ is the friction coefficient (Fig 1, supplementary). Friction is not applied on floating ice.

3. As I understand it, the authors apply a prescribed surface mass balance along with prescribed changes in terminus position to simulate changes to the dynamics of the glacier system. Moreover, the authors use an inverse method to invert for the basal friction coefficients in their sliding law. Now ice dynamics models are based on approximations to the Stokes equations. A consequence of this is that if the geometry is appropriately specified and the boundary conditions are all correctly specified the velocity is completely determined. Because the authors are using observed velocities to tune the friction coefficients and prescribing changes in terminus position, it isn't all that surprising that they can simulate the correct dynamic response. In fact, it would be surprising if the model failed given the tuning step.

We indeed do constrain the ice margin and assume that the friction coefficient $C$ does not change during the simulation, but the surface is allowed to change freely. We do not constrain the driving stress, which controls the ice dynamics. The fit between

[Figure]

**Figure 1.** Inverted basal friction coefficient

the observations and the model was therefore not necessarily expected since we perform our inversion reconstructed velocities from a first relaxation.

We re-formulate the description in the manuscript: "Thus, in the first relaxation basal friction is based on the assumption that driving stress is equal to basal stress at any given point using the initial geometry." and add Table 1 to give a better overview about the model initialisation steps.

**Table 1.** Steps for model initialisation

| Step | Input | Output |
|---|---|---|
| Relaxation 1 | GIMP extended to 1849 terminus position, Ice viscosity (initial guess), Basal friction (initial guess) | Reconstructed 1849 ice thickness and velocity |
| Thermal Inversion | Ice thickness and velocity from relaxation 1 Surface velocity from relaxation 1, Ice viscosity from thermal | Improved ice viscosity Inverted basal friction |
| Relaxation 2 | Ice thickness from relaxation 1, Ice viscosity from thermal, Basal friction from inversion | Steady state ice thickness and velocity |

What is surprising and impressive is that authors are able to get the correct dynamic response over a fairly long time interval. This seems like it is probably dictated, at least in part, by the choice of sliding law - which is why I think it is important to specify the sliding law and show us its pattern of spatial variation. Also, are the model results sensitive to the form or magnitude of the sliding coefficient. For example, do you get similar results for plastic, Coulomb and Weertman type sliding laws? How

10 sensitive are the results to the inversion? Is the good agreement a consequence of extensive model tuning or relatively insensitive to model tuning? Related to this, the authors need to be careful when comparing observations of velocity with simulated velocity. Good agreement means the inversion was able to match surface velocities, but tells us nothing about the models skill. As explained above, we apply a Coulomb-like friction law and invert for steady state velocity, not for observed ice surface

velocity.

The control run that applies only surface mass balance to the model does not show the same pattern of acceleration. The prescribed terminus position changes lead to increasing ice flow. Previous simulations to this study did not include ice viscosity retrieved from a thermal solution and others where initialised with a previous bed topography version (Morlighem et al., 2014).

5 The resulting velocity changes were the same. We conclude that the simulation results are not too sensitive to the inverted friction.

4. Related to this, I'm a bit confused by the metrics for model success. It seems to me as though the authors are comparing observations of mass balance to simulations of mass balance. However, surface mass balance is prescribed and changes in terminus position are prescribed so the only part of this that can vary is the increased dynamic discharge. Why not just compare

10 simulated change in dynamic discharge predicted by their model with that inferred from observations. For example, Figure 2 shows annual mass loss along with change in mass loss from prescribed changes in terminus position. What about also showing mass loss from prescribed surface mass balance? Then we would clearly see the component that is predicted (dynamic discharge) and what is specified.

Good idea. We modify Figure 2b to visualize the portion of prescribed mass changes to simulated total mass change and add

15 information about mass loss contributors to the text.

The more I think about it, it seems as though the observations probably give changes in trim line (or something like this) so maybe the right comparison is between predicted glacier surface elevation and observed trim lines (as opposed to ice thickness)? (I apologize to the authors if I misunderstood their data or comparisons).

Observations provide ice surface elevation from aerial photography and satellite. We only have trim line data for the little ice

20 age extent. Following advice from Referee #1 we compare ice thickness results to observations instead of ice surface elevation.

As I said before, I also don't think that the authors can claim that the match between measured velocities and observed velocities provides any test of the model. Friction has been determined by tuning the model to match observed velocities so any match between observed and simulated velocities is partly a consequence of the tuning procedure. Here I think the authors might be able to narrate to readers a bit more thoroughly what they actually measured and what they predicted (without prescribing) and

25 how the measurements can be used to test the things that the model predicted that weren't ingested in any tuning exercises.

As explained above, we invert for the relaxed velocity. Therefore, "tuning" is done for the initial state of the simulation, not for observed present day velocity. Furthermore, we compare observations to results after 160 years of simulation.

**Some miscellaneous comments**

30 Page line 13 extra space before

We could not find what you are referring to.

Don't capitalize Grounding line or Basal friction

Done.

Page 3 below 5: The ice temperature is determine by solving an advection-diffusion equation. The paper says the temperature

35 field was initialized using surface air temperature. Does this mean the ice temperature was run to steady-state using the assumed surface air temperature?

We have divided up the explanation for ice viscosity, and have moved some parts of the explanation to section 3.1, Model Initialisation. The first part remains in the Introduction section in section 3:

"Ice viscosity follows Glen's Flow law (Glen, 1955). The initial viscosity is taken from Table 3.4 in Cuffey and Paterson (2010, p. 75), assuming ice temperature of $-5°C$ and will be refined in section 3.1"

Further explanation is given in section 3.1 Model Initialisation:

"Given computed ice velocity and thickness from the first relaxation, ice viscosity and basal friction can be redefined. The ice viscosity is calculated by extruding the model with 15 layers and solving for the thermal steady state based on forcing the surface with 1854–1900 UI mean surface air temperature (Box, 2013)."

Page 3, line 10: The grounding line position is automatically calculated in each step implies that the terminus is at flotation. Why is this a good approximation and why is it forbidden for the terminus to have a thickness greater than flotation?

As explained above, the model includes a grounding line migration scheme. During the simulation, the glacier can evolve a terminus at flotation, but it is not forced to float. In most cases for our simulation, the grounding line is located directly at the terminus position.

I found a few other grammar mistakes throughout and I would urge the authors to give the manuscript one more proof read.

We improved the study regarding grammar and spelling.

**References**

Box, J. E.: Greenland ice sheet mass balance reconstruction. Part II: Surface mass balance (1840-2010), Journal of Climate, 26, doi:10.1175/JCLI-D-12-00518.1, 2013.

Cuffey, K. and Paterson, W.: The Physics of Glaciers, Elsevier Science, 2010.

Glen, J. W.: The creep of polycrystalline ice, Proceedings of the Royal Society of London A: Mathematical, Physical and Engineering Sciences, 228, 519–538, doi:10.1098/rspa.1955.0066, 1955.

Morlighem, M., Rignot, E., Mouginot, J., Seroussi, H., and Larour, E.: Deeply incised submarine glacial valleys beneath the Greenland ice sheet, Nature Geoscience, 7, 418–422, doi:10.1038/ngeo2167, 2014.

Seroussi, H., Morlighem, M., Larour, E., Rignot, E., and Khazendar, A.: Hydrostatic grounding line parameterization in ice sheet models, The Cryosphere Discussions, 8, 3335–3365, doi:10.5194/tcd-8-3335-2014, 2014.

---

## Author Comment (AC3) · 12 Oct 2017

[revised manuscript text omitted]

*Correspondence to: Konstanze Haubner (khu@geus.dk)*

**Model initialisation**

[Figure]

**Figure 1.** Inverted basal friction coefficient

**Simulation comparison: 1849 to 2012**

[Figure]

**Figure 2.** Simulated ice thickness and velocity changes 2012–1849.

**Ice thickness comparison: simulation - observations**

[Figure]

**Figure 3.** Comparison of simulated and observed ice thickness in 1985 (Korsgaard DEM), 2005 (GIMP) and 2012 (ArcticDEM).

**Surface velocity comparison: simulation - observations**

[Figure]

**Figure 4.** Comparison of simulated and observed ice surface velocity in 1991/1992 (CCI, Envisat), 2000/2001 (MEaSUREs) and 2009/2010 (MEaSUREs), representing begin,middle and end of observation time.

**Mass change comparison - GRACE**

[Figure]

**Figure 5.** (a) GRACE area overview. UI catchment and model domain (blue polygon), present Greenland ice margin (green line) and the GRACE mascon (red line). (b) Mass change comparison between 2003 and 2012. GRACE (red), simulated mass loss of the intersection area of mascon 123 and model domain (dashed line) and a simulated mass loss (sum of ISSM_PT simulated mass loss in mascon 123 and the ISSM SSA model output of Schlegel et al. (2016) to cover the entire domain) (blue line).

**References**

Schlegel, N.-J., Wiese, D. N., Larour, E. Y., Watkins, M. M., Box, J. E., Fettweis, X., and van den Broeke, M. R.: Application of GRACE to the assessment of model-based estimates of monthly Greenland Ice Sheet mass balance (2003-2012), The Cryosphere, 10, 1965–1989, doi:10.5194/tc-10-1965-2016, 2016.

---

## Author Response (AR2)

**Interactive response to reviewer comments on* "Simulating ice thickness and velocity evolution of Upernavik Isstrøm 1849–2012 by forcing prescribed terminus positions in ISSM"**

**by Konstanze Haubner and co-authors**

We thank both reviewers for their constructive comments on our manuscript. We feel the requested changes have improved the clarity of the paper and appreciate their feedback. The author response to reviewers is structured as follows:

– Reviewers' comments in blue

– Authors' response in black

We significantly rewrote some sections of the manuscript, to improve clarity. The major changes include:

– updated comparison of ice mass changes in section 4.2

– included additional figure in supplementary, visualizing simulated frontal changes over time

– supplementary figures comparing observations and simulation results focus now on the frontal area (zoom)

– additional paragraph in the discussion, explaining limitations of the here described method regarding mesh and time steps

Our conclusions remain unchanged. The reviewed manuscript with tracked changes is attached.

**Referee #1 (anonymous)**

**General comments**

The authors have significantly improved the paper by clarifying various aspects of the methods and providing more detail and new figures with respect to model performance. The paper should be published subject to minor revision.

The authors suggest that it's inappropriate to distinguish between prescribed and resultant dynamic mass loss when comparing with observations, and that the methodology is motivated by that of Khan et al. (2013) and Larsen et al. (2016). Examining these papers more closely, it seems that the mass loss estimate of Khan et al. (2013) excludes changes in terminus position, while Larsen et al. (2016) include the terminus retreat (I believe), but also provide data on partitioned loss (dynamic loss upstream and downstream of flux gates near the terminus). Can the authors confirm that the simulation data is comparable?

Yes, we confirm that the data is comparable.

Khan et al. (2013) excludes the area of retreat from the mass change calculations and states "Taking into account that this is a minimum mass loss rate because the area of retreat is not included when computing the volume loss, we estimate a minimum catchment-wide, dynamically induced mass loss of $72.3 \pm 15.8$ Gt during 1985 to 2010." We conclude, that Khan et al. (2013) would include the volume loss from the area change into the dynamically induced loss, if they had a good measure for it. The

5   simulated mass changes that are compared to Khan et al. (2013) are extracted from the simulation, considering the catchment area used in Khan et al. (2013), keeping the ice margin fixed.

Larsen et al. (2016) include the ice volume corresponding to the area of retreat in their dynamical induced mass changes, e.g. they define in equation (2): $DIL = \Delta F + (\Delta V - \Delta SMB)$, where $\Delta F$ is the change in flux through the fluxgate and $\Delta V$ change in ice volume, downstream of the fluxgate. We calculate ice mass changes for comparison on the catchment defined by

10   Larsen et al. (2016) and include frontal changes.

P3 L13: Equation 1 refers to effective pressure. How is this defined inland? Does it remain fixed through time?

Equation 1 is defined for all grounded ice. Hence including ice further inland.

We improve the sentence referring to the equation 1 to:

*A Coulomb-like friction law is applied on **all** grounded ice:*

15   The effective pressure does not stay fixed through time. Moreover, it is updated given the new ice thickness at every time step. We change the sentence after the equation to:

*The effective pressure $N$ is updated at every simulation time step, given the new ice thickness. The friction coefficient $C$ is variable in space, but constant in time (Fig 1, supplementary).*

20   **Specific Comments**

Supplementary Figure 1 has no units or scale bar

We added a scale bar and unit for the friction coefficient.

Supplementary Figure 2 – misspelled 'relative'

Corrected.

**Referee #3 (anonymous)**

**General comments**

The paper submitted provides an evaluation of thickness and velocity evolution for Upernavik Isstrøm for two sets of simulations: (1) driven entirely by SMB; and (2) driven by SMB and prescribed changes in calving front location, thus circumventing the need for a calving law. The authors find that the present day velocities are generally replicated by the end of the simulation driven primarily by calving front change. If valid (see below), this is impressive given that the transient part of the model simulation time is 160yr.

Having read through the previous version of the manuscript and sets of reviewer comments, the paper is greatly improved in terms of clarity regarding model initialisation. However, I have a few concerns regarding the interpretation of some of the model/observed comparison measures, in addition to a couple of questions for the authors that could be addressed by two/three more model simulations and allow for a better evaluation of their conclusions.

**Prescription of calving in the model**

Rather than implement a calving law, the authors prescribe terminus change based on their observational record which spans from 1849-2012. Their prescription of calving assumes that terminus change from the previous observation occurs either instantaneously or (for larger calving events) via staged retreat between observations evenly spread through time. Given that one of the author's main conclusions is that terminus change strongly impacts dynamics, for me there is currently a large gap in the paper in testing the sensitivity of the final results to the manner and timing of how the retreat is prescribed. For me, this gap currently casts doubt on the author's argument in the abstract that the results of their terminus change simulation could be used as a metric for evaluating the effect of calving laws.

To take an example, the terminus change between any two successive observations could have occurred extremely rapidly, or it could have occurred more gradually. If this was to be investigated by the model there are two extremes of how terminus change is prescribed: (1) instantaneously forcing retreat of the calving front irrespective of the distance of retreat (admittedly highly unrealistic); and (2) forcing retreat between every terminus observation in steps, similar to how the larger calving events are dealt with in the existing simulation. Within the second scenario there is also likely to be sensitivity to how often the calving front is updated (e.g. every timestep or every 6 model months say).

My understanding from the paper is that the authors have taken a compromise approach between scenarios 1 and 2 in order to provide a more 'realistic' simulation, and are able to identify 0.5-6 month accelerations that are coincident with prescribed retreat events. By investigating scenarios 1 and 2 outlined above it could be evaluated whether the dynamics and geometry after velocities have stabilised are conditioned by the terminus change events themselves, or just the terminus position. This is important to know if the results of this paper's simulations are going to be compared to model results where calving laws have been implemented (i.e. for decadal-centennial timescales is it important to regularly update your calving front location, or will you get different/similar decadal timescale dynamics (and ice volume lost) by updating at intervals of every few years?).

The here discussed method can not be applied gradually and is therefore limited to stepwise prescribed retreat, which is defined by the observation times, the observed terminus positions and the mesh resolution. Including additional terminus positions is

restricted to the space in between the observed front positions and the mesh resolution and only few adaptions can be made in the setup of this publication. Hence, we run one test without intermediary positions, only prescribing the actual observed frontal changes (see Fig. 1). We include Fig. 1 and 2 into the supplementary and add the following paragraph into the discussion:

*Simulated frontal changes occur on annual time scales, marked by observation years, and happen thereby less often than in*

5    *nature. In addition, the magnitude of the removed ice mass is defined by the mesh resolution, set between 300 and 800 m on the area of frontal retreat. Hence, simulated frontal changes appear more abrupt than in nature and do rather capture changes in glacier velocity and thickness on decadal time scales than a seasonal pattern. The simulation is likely to overestimate the velocity and thickness changes in response to the larger decrease in frontal backstress.*

*Moreover, the timing of the prescribed terminus changes of ISSM$_{PT}$ is not well constrained, given by observations with gaps*

10    *of up to 13 years after 1900. In ISSM$_{PT}$, we include 20 additional calving fronts, to minimize the ice mass removed at each terminus change. A simulation excluding the 20 interpolated terminus positions does not affect the overall simulation results (see supplementary). The total ice mass change shows higher peaks, but results in similar cumulative mass changes. Simulated 2012 ice velocity and thickness of ISSM$_{PT}$ and the sensitivity simulation agree as well with $\pm5\,m\,yr^{-1}$ and $\pm3\,m$ respectively. However, given more frequent observations could minimize the timing uncertainties and with multiple observed terminus front*

15    *positions throughout a year, this approach could capture seasonal glacier changes.*

[Figure]

**Figure 1.** Annual ice mass changes from ISSM$_{PT}$ (blue) and a test simulation (red) that is sole prescribed with observed front positions to estimate sensitivity of the prescribed-terminus change method to intermediate prescribed frontal changes. Prescribed termini changes are marked with dashed (observations) and dotted (interpolations) lines.

     We reformulate the statement how this method can be used in the conclusion (p13, l. 33) to:

*With an increasing amount of collected observed terminus front positions, the here discussed method will become a progressively more useful tool to* evaluating calving laws or determining calving law parameters for hind-cast simulations before they

20    are applied to future simulations. [...]

**Comparison of model results with observations**

I'm not sure that the author's statement on p11 L12 that "In 1985–2012 ISSM$_{PT}$ simulates mass changes similar to observations" is fully justified/qualified (this is of course heavily dependent on what you mean by "similar"). Looking at Table 3, only

[Figure]

**Figure 2.** Spatial difference between ISSM_PT and a test simulation (red) that is sole prescribed with observed front positions.

1985-2002/5 mass and dynamic changes represent a good match with the Kahn et al. (2013) values, and 2009-11 of the Larsen et al. (2016) values. The remainder of the values quoted in some cases have substantial mismatches even when the full error range is taken into account. The authors need to be a bit more up front about this, but also critique the reasons why this may be the case (e.g. the length of observation versus the fact that these comparisons are being made for the end part of the transient
5     run). Personally I think getting the good match between obs and modelled results for 1985-2002/5 represents a fantastic result in itself, though the reasons for the mismatch at other times needs to be explored and explained.

We agree and admit, that we focussed here too much on the relative comparison of the DIL contribution to total mass changes instead of absolute values. We change the statement to the period 1985–2002/2005 and address the later mismatch between 2002/2005–2009 by saying:

10    *In 1985–2002/2005, ISSM_PT simulates mass changes similar to observations (Kjær et al., 2012; Khan et al., 2013; Larsen et al., 2016). However, observed and simulated mass changes between 2000 and 2008 differ from each other showing up to 50% more simulated than observed total mass loss (Table 3). During this time, the largest area changes are prescribed at UI-1 with increasing retreat rates from 200–500 m y^{-1} between 1985 and 2005 to 4 km y^{-1} in 2006–2009 along the UI-1 centre flow line. Between 2006 and 2007, UI-1 splits into three calving fronts, followed by the disintegration of UI-1's floating ice tongue*
15    *in 2008 (Larsen et al., 2016). The simulated UI-1 terminus is however grounded between 1990 and 2012, only starting to float above an overdeepening in the bathymetry in 2007 in order to be grounded again after the prescribed retreat in 2008. The misfit between observed and simulated mass change could be justified by the absent floating tongue in the simulation. The UI-1 bathymetry is deeper than 500 m below sea level. Therefore, prescribed retreat leads to higher ice mass loss retreating over the same distance than an observed disintegration of a floating tongue. Moreover, a floating tongue has potential to stabilize the*
20    *glacier more, decreasing the glacier velocity and dynamical discharge Nick et al. (2012).*

It's also unclear at times in the manuscript what numbers quoted actually represent. For example, P9 L16 quotes the surface as lying within ±30 % of the observed thickness, though it is not immediately obvious whether this is for the lower portion of the catchment affected dynamic drawdown or for the entire catchment. Given that a lot of the behaviour is driven by calving,

a much more detailed view of the near-terminus region would be useful throughout the supplementary (especially though for figure S3) as it would provide a much better impression visually of what is described in the text in addition to where the mismatches are occurring.

We agree to set more focus to the near-terminus area when comparing simulation results and observations and include additional figures for the comparison with ice velocity and thickness in the supplementary. The temporal comparison of initial and final simulation results is replaced with the more detailed view on the changes close to the 2012 ice margin.

**Minor points**

P5 L6 – grammar could do with clarifying

We change the sentence to: "Over the present day ice covered area, the initial ice surface is given by the 2005 ice surface elevation"

P5 L10 – how was the ice surface interpolated and how was the post-relaxation 1849 model configuration compared with the reconstructed configuration

The surface elevation is interpolated linearly between the given trimline points, GIMP and ice margin. We adjust the text and include an additional figure in the supplementary:

*In the remaining area the ice surface elevation is interpolated linearly being constraint to a minimum elevation of 40 m. [...] During the relaxation, the reconstructed glacier area thickened by 50–400 m from the UI-1/UI-2 1966 terminus position reaching 40 km upstream, while the ice surface velocity slowed down by a maximum of 2.5 km $y^{-1}$. Along UI-3, the glacier thickness changed by $\pm150$ m and ice surface velocity decreased by 500–1500 m $y^{-1}$ (see supplementary).*

P6 L17 – would remove the link between air T's and the likelihood of retreat unless it can be fully substantiated

We understand your concern, though we would like to explain the choice of prescribed terminus position change and include to references, linking increased submarine melt and glacier retreat to increased meltwater runoff:

*The highest surface air temperatures and melt rates on UI are observed in July (van As et al., 2016), increasing the likelihood of terminus retreat (Sciascia et al., 2013; Fried et al., 2015).*

P6 L19-20 – explain the time steps of calving fronts that are used

We chose the time steps of calving fronts to be either the observation time or halfway points in time between observations. The sentence at line 19/20 is changed to:

*The additional calving fronts are prescribed at the halfway points in time between observations and aim to improve realistic simulation behaviour by splitting 20 large ice area changes induced by the prescribed terminus changes into smaller areas within shorter time periods (dotted lines, Fig 3).*

P6 L22 – spell out CFL

Done (Courant-Friedrich-Lewy condition).

P7 L5-6 – if simulations are run with stepwise/gradual prescribed calving retreat, is the average surface vel increase the same as the original simulation?

As described above, we do not run additional simulations with gradual prescribed retreat.

Prescribing the simulation sole with observed frontal positions and excluding the intermediary interpolated prescribed retreats does not change the average surface velocity and thickness changes (see Fig. 2).

P7 L11 – replace "succeeding" with "following"

Done.

P9 L2-3 – I assume the observational data %s aren't exactly the same as the modelled DIL values – the observed percentages are needed for comparison.

We included the percentages of DIL from total mass change (Table 3).

P9 L9-12 – in addition to the movies can you provide a figure showing time evolving evolution of the surface (similar to Jamieson et al., 2012, , figure 2). Personally, I find it much easier to visualise and interpret glacier change this way rather than having to play a movie over and over.

Of course. Figure 3 is included in the supplementary.

[Figure]

**Figure 3.** Glacier surface elevation along the centre flow lines og UI-1, UI-2 and UI-3 shown at 5 y intervals over the 163 y simulation period. The grey areas show the bed geometry. Prescribed termini changes are marked with dashed (observations) and dotted (interpolations) lines.

P10 L11 – for which locations on UI1,UI2 and UI3 are these velocity increases measured at? Are they the same fixed point or a given distance upstream of the terminus?

These velocity increases are taken as an average along the flow lines between 1 km and 30 km upstream the 2012 terminus. We adjust the sentence accordingly:

*By the end of the ISSM$_{PT}$ simulation, ice flow velocity has doubled at UI-1 and UI-2 and increased by 55 % at UI-3 compared to 1849 (relating the 1849 and 2012 velocity along each flow line between 1 and 30 km upstream the 2012 terminus position).*

P11 – make axes on figure 5 equal

The axes are equal. We adapted the ticks on y- and x-axis to avoid confusion.

P12 L26 – how does the model deal with advance? Is it mass conserving or is mass in effect added to the domain?

Ice mass is advected downstream and extrapolated over the regions that are newly activated. We add in the text:

*When ice margin advance is prescribed, ice mass is advected downstream and extrapolated over the regions that are newly activated.*

[revised manuscript text omitted]

---

## Author Response (AR4)

**Response to editors minor review on* "Simulating ice thickness and velocity evolution of Upernavik Isstrøm 1849–2012 by forcing prescribed terminus positions in ISSM"**

**by Konstanze Haubner and co-authors**

Dear Olivier,

Thank you for accepting this manuscript for publication in The Cryosphere!

To address your last comment, I changed the sentence defining the effective pressure. The water pressure is not null everywhere. The $h$ in the equation is the ice thickness above floatation, not the ice thickness itself.

5    In order to clarify, the effective pressure is now expressed as a combination of ice thickness and the bedrock elevation:

*The effective pressure $N$ is assumed to be $N = g\left(\rho_i\, H(x,y,t) + \rho_w\, b(x,y)\right)$ where $H$ is the ice thickness at the current time step, $b$ is the bed elevation with respect to sea level, $g$ is the gravity, $\rho_i$ and $\rho_w$ are the densities of ice and water respectively.*

Thanks again for your work and effort to improve this manuscript.

10    The reviewed manuscript with tracked changes is attached.

[revised manuscript text omitted]